

# Accounting for model error in air-quality forecasts: an application of 4DEnVar to the assimilation of atmospheric composition using QG-Chem 1.0

Emanuele Emili[1], Selime Gürol[1], and Daniel Cariolle[1]

[1]CERFACS

*Correspondence to:* Emili (emili@cerfacs.fr)

**Abstract.** Model errors play a significant role in air-quality forecasts. Accounting for them in the data assimilation (DA) procedures is decisive to obtain improved forecasts. We address this issue using a reduced-order chemical transport model based on quasi-geostrophic dynamics and a detailed tropospheric chemistry mechanism, which we name QG-Chem. This model has been coupled to a generic software library for data assimilation and used to assess the potential of the 4DEnVar algorithm for air-quality analyses and forecasts. Among the assets of 4DEnVar, we reckon the possibility to deal with multivariate aspects of atmospheric chemistry and to account for model errors of generic type. A simple diagnostic procedure for detecting model errors is proposed, based on the 4DEnVar analysis and one additional model forecast. A large number of idealized data assimilation experiments are shown for several chemical species of relevance for air-quality forecasts ($O_3$, $NO_x$, CO and $CO_2$), with very different atmospheric life-times and chemical couplings. Experiments are done both under a perfect model hypothesis and including model error through perturbation of surface chemical emissions, for two meteorological and chemical regimes. Some key elements of the 4DEnVar algorithm such as the ensemble size and localization are also discussed. A comparison with results of 3D-Var, widely used in operational centers, shows that, for some species, analysys and next day forecast errors can be halved when model error is taken in account. This result was obtained using a small ensemble size, which remain affordable for most operational centers. We conclude that 4DEnVar has a promising potential for operational air-quality models. We finally highlight areas that deserve further research for applying 4DEnVar to large scale chemistry models, i.e. localization techniques, propagation of analysis covariance between DA cycles and treatment for chemical non-linearities. QG-Chem provides a useful tool in this regard.

## 1 Introduction

In recent years, Data Assimilation (DA) of atmospheric constituents has become a key tool for providing more accurate forecasts and reanalyses of the atmospheric composition. The increasing availability of chemical observations from both satellites and ground-based instruments allowed to reduce the uncertainty of atmospheric chemistry models in a large number of applications. Utilization of DA can be found in the modeling of volcanic ashes (Lu et al., 2016), in operational air-quality forecasts at continental scale (Marécal et al., 2015) or in the reanalysis of the global atmospheric composition at decennial scale (van der A et al., 2010; Inness et al., 2013). Data assimilation can also be used to infer surface fluxes of long-lived chemical compounds





(Thompson and Stohl, 2014; Chevallier et al., 2005). A review of the utilization of data assimilation for the atmospheric composition can be found in Zhang et al. (2012); Bocquet et al. (2015).

The main goal of DA is to reduce the uncertainties of a model through a timely combination of model results and observations. This is generally done by means of correcting the so-called *control variables* of the given model. The choice of the
control variables should reflect the largest source of uncertainty of the considered model. In atmospheric chemistry control variables are typically associated with the model initial state (Elbern et al., 1997) or chemical emissions (Chevallier et al., 2005). However, inaccurate identification of the model uncertainty can lead to a wrong adjustment of the model through DA, even though the spread between model predictions and assimilated observations is reduced (Tang et al., 2015). The choice of the control variable, and the approximate knowledge of its uncertainty (also named background error covariance), is therefore
critical for the design of an appropriate assimilation algorithm and to ensure correct results of DA.

Principal sources of uncertainty of atmospheric chemistry models include the model initial condition, model ancillary data or parameters, model physical parametrization, chemical mechanism etc. (Beekmann and Derognat, 2003; Mallet and Sportisse, 2006). Since different chemical species can be sensitive to different physical and chemical processes, the main sources of uncertainties can also differ from species to species. For example, long-lived species like $CO_2$ or CO are mostly sensitive
to uncertainties in surface fluxes, which can be corrected using variational algorithms in combination with long assimilation windows of several weeks (Chevallier et al., 2005; Koohkan and Bocquet, 2012). This is possible since the chemical reactivity and the sensitivity to the initial condition are negligible for long integration of the model. Uncertainty in the transport processes is also generally neglected (Babenhauserheide et al., 2015). Data assimilation of short-lived gases like tropospheric $O_3$ or $NO_2$, which is encountered in air-quality applications, is instead trickier. $O_3$ and $NO_2$ are involved into rapid chemical reactions and
sensitive to several model parameters ranging from reaction rates, emissions of primary species, clouds and radiation, boundary layer mixing etc. It is much more difficult in this case to identify a single and predominant source of uncertainty in the model predictions.

In most air-quality operational models a pragmatic choice is currently done by setting the control variable to the initial state of the measured species and using short forecast/assimilation cycles (e.g. 1 hour). Since ground-based measurements are
generally available at hourly frequency, most used sequential DA algorithms such as Optimal Interpolation (OI), 3D-Var or Ensemble Kalman Filters (EnKF) (Marécal et al., 2015) all provide a strong constraint on model trajectories (Wu et al., 2008). This strategy gives robust results for operational analyses because chemical fields are corrected every hour. Since the control variable correspond to the assimilated observations, this also permits to estimate the background error covariance from previous validation of the model against observations (Hollingsworth and Loennberg, 1986), and keeps from the difficult diagnosis of
the true model uncertainties.

However, the model dynamics is neglected in OI or 3D-Var DA schemes. Attempts of using ensembles of model analyses to specify dynamically the background error covariance within 3D-Var did not show clear improvements over the static case (Jaumouillé et al., 2012). Specification of cross-correlations between interacting chemical species in the 3D-Var background error covariance matrix is also particularly difficult, because chemical interactions depend on the local concentrations and on
meteorological conditions. As a consequence, multivariate chemical DA with 3D-Var schemes has not been yet documented



in the literature. In EnKF systems, the forecast model is used to propagate and estimate the background error covariance, but ad-hoc adjustments are necessary to avoid the collapse of the ensemble variance and obtain realistic covariance matrices for 1 hour forecasts (Gaubert et al., 2014; Constantinescu et al., 2007a). As a result, costly algorithms such as EnKF or 3D-Var hardly give better results that more simple OI for chemical reanalyses (Rouil and the MACC team, 2014). More important,

very little improvement is obtained, whatever the employed DA algorithm, for the next day model forecast (Wu et al., 2008). next day forecasts of reactive gases such as $O_3$ or $NO_2$, but also other other pollutants such as aerosols mixtures (PM10 and PM2.5), depend weakly on the initial condition and are more sensitive to model settings such as surface emissions or physical parametrizations. Current operational systems can achieve accurate reanalyses of observed chemical species through DA but parameter estimation and, more in general, model errors must be taken in account in DA to improve chemical forecasts of

reactive gases and particles.

Some studies evaluated more advanced DA algorithms to jointly correct surface emissions of precursor species and initial condition of observed species. For example, Elbern et al. (2007) employed a 4D-Var scheme in combination with assimilation windows of 24 hours to assimilate $O_3$, $NO_x$ and $SO_2$ measurements. A similar study has been done also in the context of a toy-model experiment by Hamer et al. (2015), where only emissions of precursor species are adjusted to improve $O_3$ forecasts.

Results seems promising but still relies on the assumption that the model is *perfect*, i.e. that there are no additional sources of uncertainties in the model forecast other than the controlled variables (i.e. the initial state and the selected emissions). This can lead to the over-correction of control variables when non negligible model errors exist. Concerning EnKF implementations, some authors also tested joint optimization of the chemical state and precursor emissions (Miyazaki et al., 2012; Tang et al., 2011; Constantinescu et al., 2007b). EnKF includes naturally model uncertainties in its formulation, which can be added

through stochastic perturbation of model parameters during the ensemble forecast (Evensen, 2003). However, EnKF corrects the model trajectories sequentially. In a typical air-quality context, the emissions of $O_3$ precursor species (e.g. NOx and VOCs) in the early morning or night can affect the concentration of observed species (e.g. $O_3$) in the early afternoon, when the photo-chemistry takes place. When using EnKF, the information made available by afternoon measurements cannot be used to correct the model at previous hours.

Based on above-mentioned facts, we reckon that the following capabilities are needed to further improve DA in air-quality applications:

- permit simultaneous assimilation and optimization of multiple chemical species (multivariate DA), with possible chemical interactions and very different life-times;

- include model error and account for disparate sources of model uncertainty;

- allow long assimilation windows to make best use of the information content of frequent air-quality observations.

This can be accomplished by using the so-called *weak constraint* 4D-Var (Tremolet, 2006), which is an extension of the 4D-Var algorithm that accounts for the model error. On top of the operators already needed to perform the strong constraint 4D-Var (e.g. the tangent linear and adjoint codes of the forecast model), the formulation of the weak constraint 4D-Var requires the



definition of the model error covariances $\mathbf{Q}$, which can be difficult to estimate in real applications (Trémolet, 2007). Recent studies have shown that the linear operators needed in the weak constraint 4D-Var formulation can be approximated through an ensemble of model forecasts. For example, the 4D-Var-EnKS (Mandel et al., 2016) uses an ensemble approach to mimic the tangent linear and adjoint model for the minimization of the weak constraint 4D-Var cost function. The IEnKS (Bocquet and

Sakov, 2014) is a fixed-lag ensemble Kalman smoother formulated under perfect model assumptions, which can also be used to estimate erroneous model parameters through an augmented state formalism (Haussaire and Bocquet, 2016). The 4DEnVar method (Desroziers et al., 2014) uses an ensemble of non-linear model trajectories to estimate both the error covariances for the initial condition and the model error, as well as to approximate the tangent linear and adjoint model. These type of approaches are generally referred in the literature as ensemble-variational EnVar (Lorenc, 2013), as opposed to 'hybrid' methods, which

make use of ensembles only to specify error covariances matrices in variational algorithms (Belo Pereira and Berre, 2006). EnVar methods have a major advantage for atmospheric chemistry applications: they avoid the construction of tangent linear and adjoint codes of the forecast model, which still lack for most of the operational CTMs, or are becoming very difficult to be maintained due to the rapid evolution of models and computer architectures.

The main advantage of the 4DEnVar method is that it permits to account for a generic model error through the addition of

stochastic perturbations during the model integration step (like in EnKF). Moreover, it focuses exclusively on the estimation of the model state, which is the only variable that is directly constrained by observations. This avoids on one hand the difficult specification of $\mathbf{Q}$ still needed in the 4D-Var-EnKS and, on the other hand, the need in IEnKS to select a number of model parameters among all the possible erroneous parameters in complex CTMs. As all ensemble based methods, 4DEnVar also supports naturally multivariate chemical DA, with the cross-covariances terms between chemical species being automatically

obtained from the ensemble of the non-linear model forecasts.

Variants of the 4DEnVar have been already tested in real numerical weather prediction (NWP) applications (Lorenc et al., 2015). The method has proven to be affordable for large scale operational NWP models, even though the skills of the operational hybrid 4D-Var are not yet matched. To the knowledge of the authors, EnVar type methods have not yet been implemented in air-quality or atmospheric chemistry models and no previous study has already examined the potential of 4DEnVar for chemical

DA. Note that, operational NWPs are already based on well mature 4D-Var DA systems, whereas very few atmospheric chemistry models do. Therefore, there is a larger room for improvements from EnVar type of algorithms in air-quality models then in NWP. Hence, the main objectives of this study are:

- to present a new atmospheric chemistry toy-model built for assessing and comparing performances of several DA algorithms;

- examine the potential and limits of the 4DEnVar algorithm for air-quality analyses, compared to the generally used 3D-Var;

- present a new procedure based on 4DEnVar to improve chemical forecasts on the next day.

The purpose is to examine state-of-the-art DA algorithms in the reactive-gases/air-quality context and, therefore, to guide future developments for the operational DA systems. Four gaseous species with very different life-times and chemical mechanisms,





currently well observed either from satellites or from ground-based instruments, are considered for this study ($CO$, $O_3$, $NO_2$ and $CO_2$). Using a simplified model in this context permits: faster implementation of complex DA algorithms, cheaper numerical experiments and more straightforward interpretation of the DA results (Fairbairn et al., 2013). The latter is particularly true compared to DA experiments done using real observations, with generally unknown error statistics. Compared to already
mentioned simplified models (Hamer et al., 2015; Haussaire and Bocquet, 2016), which are respectively 0-D and 1-D, the newly proposed model is 3-D and uses the same tropospheric chemistry scheme of operational air-quality models. This allow to reproduce more features of real models, for example the complex interactions of reactive chemistry and large scale advection or the effect of boundary conditions. This also permits to better examine typical issues of DA within large systems, like the emergence of sampling errors due to the finite size of the ensemble and the consequences of localization techniques.
Additionally, the use of 3-D fields and operators eases the estimation of numerical costs and possible bottlenecks of the DA algorithm in terms of operational implementation. We remind that the objective of this study is to demonstrate the applicability of a DA algorithm that outperforms currently implemented methods in operational centers, but with an acceptable compromise between computational costs and precision. Finally, the toy system has been implemented under a generic library for data assimilation, to ease the exchange of assimilation algorithms or toy-models between scientists.
15       The paper is outlined as follows. The developed atmospheric chemistry model will be presented in Section 2. A summary of the data assimilation algorithms employed in this study is given in Sec. 3. The first section of the numerical results (4.1) presents a number of DA experiments done under the hypothesis of perfect model. A detailed comparison between 4DEnVar and 3D-Var is presented, as well as a sensitivity study on the principal parameters of the 4DEnVar algorithm, i.e. the ensemble size and the localization choices. In the following Section (4.2) the effects of a model error is investigated using the 4DEnVar
algorithm. A statistical comparison of the 4DEnVar and 3D-Var performances on multiple cycles of analyses and forecasts is presented in Sec. 4.3. Finally, conclusions are given in Sec. 5.

## 2   Model description

A new atmospheric chemistry low-order model has been developed for this study and named QG-Chem. The objective was to reproduce typical features of chemical fields from large scale Chemical Transport Models (CTM), but maintaining the
computational cost low enough to allow a comfortable usage on a personal computer. The meteorological forcing is computed using a 2-layers Quasi-Geostrophic (QG) model, representative of mid-latitudes mesoscale dynamics (Pedlosky, 1992). The QG wind field is used to advect the chemical species, which makes QG-Chem a coupled meteorological-chemistry model. This choice permits to examine the behavior of DA in presence of complex gradients of wind fields and vorticity. Since all DA algorithms make strong assumptions on the model dynamics (Sec. 3), it is important to test them in presence of advection
patterns that can be found in real applications. Nevertheless, the focus of this study remains atmospheric chemistry. Therefore, a detailed tropospheric chemical mechanism has been considered. The details of the meteorological and chemical models are given next.



## 2.1 Quasi-geostrophic meteorology

The 2-layers QG model is a geophysical fluid model composed of two atmospheric layers of fixed depth and potential temperature. It is a simple model of the atmosphere at mid-latitudes, whose main forcings are represented by the Coriolis force and the orography or surface heating. The governing equation is the conservation

$$\frac{\mathrm{D}q_i}{\mathrm{D}t} = 0 \tag{1}$$

of the potential vorticity $q = (q_1, q_2)$ expressed in non-dimensional variables (Fandry and Leslie, 1984):

$$q_1 = \nabla^2 \psi_1 - F_1(\psi_1 - \psi_2) + \beta y \tag{2}$$

$$q_2 = \nabla^2 \psi_2 - F_2(\psi_2 - \psi_1) + \beta y + R_s \tag{3}$$

where the subscripts 1 and 2 stand for the top and bottom layer respectively, $\nabla^2$ is the two-dimensional Laplacian, $R_s$ represents orography or heating, $\beta$ is the (non-dimensionalised) northward variation of the Coriolis parameter at a fixed latitude, $F_1$ and $F_2$ couple the layers together being a function of Coriolis force, layer depths, gravity and typical length scale. The 3-D streamfunction $\psi = (\psi_1, \psi_2)$, whose horizontal derivatives give the horizontal wind field $u_i, v_i$, can be considered as the model state vector.

The sources of the QG model provided by ECMWF have been used for this study (Y. Trémolet, personal communication). The depth of the two layers, the resolution of the horizontal grid and the integration time step $\Delta t$ are the main model parameters that can be set at run-time. The dimensional scaling and model orography are fixed, as well as the extension of the domain, which is 12000 km in the zonal direction and 6300 km in the meridional direction. The boundary conditions are taken cyclic in the eastward direction and are fixed to climatological values at meridional walls. For all the experiments presented in this study a coarse resolution of 16x8 grid points has been used. We remind that the focus of this study is to test chemical DA algorithms in a toy-model framework. Therefore, there are no stringent requirements on the realism of the meteorological fields and no need to reproduce a real atmospheric situation. The only desired property is to obtain wind fields that exhibit typical patterns of the complex atmospheric circulation. A summary of the QG model parameters used in this study is detailed in Table 1.

## 2.2 Tropospheric chemistry

The state vector of the QG model has been extended to include chemical species. The Regional Atmospheric Chemical Mechanism RACM (Stockwell et al., 1997), which describes 96 chemical species with about 300 reactions, has been implemented. This chemical scheme has been developed for air-quality modeling and is currently used by a number of operational models in Europe (Marécal et al., 2015). Photo-chemistry and its diurnal cycle are included via look-up tables, assuming global clear-sky conditions. Surface fluxes of chemical species are assigned at run-time and are kept constant during the temporal integration of the model. Chemical species are advected by the QG wind field using the semi-lagrangian scheme used for solving the QG governing equation (1).

After the advection of the species, their concentrations are updated by addition of the chemical tendencies. They are computed by solving the stiff ODE system that describes the adopted chemical mechanism. The ODE system is of the non-linear





form:

$$\partial C/\partial t = f(C) = P(C) - L(C) \cdot C \qquad (4)$$

where $C$ represents the local species concentrations, $P(C)$ and $L(C)$ the production and loss terms. The stiffness of the systems comes from the wide range of values that can take the loss terms leading to large range of chemical lifetimes.

The above system is integrated using the Adaptive Semi-Implicit Scheme ASIS (Cariolle D, personal communication). ASIS is a one-step semi-implicit scheme with prognostic time step which has a special treatment of the Jacobian matrix that diminishes the stiffness of the system. To solve the system of linear equations associated to the semi-implicit scheme, ASIS uses the generalized minimal residual method (Saad and Schultz, 1986) which appears to be very competitive in terms of computer time with good convergence. The ASIS solver is mass conservative and adapts its sub-timestep to the adopted tolerance errors
as described by Verwer (1994). In our application the ASIS solver uses a 7 s minimum sub-timestep, an absolute error tolerance of $10^4$ molecules cm$^{-3}$ and a relative tolerance error of 0.01. Therefore, a common integration time-step for both the dynamical and chemical solvers is used, which is set to 10 minutes to ensure reliable chemistry solutions.

The meridional boundary conditions for chemical species are set to climatological values. The values of surface pressure and temperature used by the chemical mechanism are fixed globally and do not depend on the QG fields. Moreover, no physical
removal process for the chemical species has been included in the model so far. These modeling choices let this study focus on the following main processes of air-quality models: emissions, chemistry and transport. Therefore, only the bottom layer of the QG-Chem model will be analyzed throughout this study. A summary of the chemical configuration is given in Table 1.

### 2.3 Description of the case studies

We considered two type of atmospheric situations in this study, to examine the performances of DA in the following conditions:

1. weak zonal winds and relatively clean atmosphere

2. developed Rossby waves and polluted atmosphere

The two situations are obtained using the same model configuration, initial condition (Tab. 2) and surface emissions, but letting the spin-up period last respectively 1 and 20 days. Emissions (Tab. 3) are taken from the study of Crassier et al. (2000), FLUX case, representative of the urban environment of Paris. Spatial heterogeneity of chemical concentrations is obtained by scaling
the reference emissions in Tab. 3 by 0.01, 1, 0.25 respectively on the western, central, eastern parts of the domain (Fig. 1). This permits to evaluate DA in different chemical regimes and when air-masses are transported between polluted and clean regions, as occurs in real applications. The three regions can be imagined as a clean marine region, a urbanized area and a rural one.

Since no chemical removal process is included in QG-Chem, long-lived species that are emitted or produced by chemical reactions of precursor species, are allowed to grow in time. The chemical concentrations at the meridional boundaries are set
to the same values as in Tab. 2. Hence, the presence of meridional boundaries can eventually counterbalance the growth of long-lived species by advection of clean air-masses from outside the domain.





**Table 1.** QG-Chem model parameters and non-dimensional scaling factors. The parameters marked by * are fixed globally and only relevant for the chemical mechanism.

| Characteristic | Description |
|---|---|
| Geographical domain | 12000 km (E-W) x |
| | 6300 km (N-S) |
| Zonal resolution (grid points) | 750 km (16 grid points) |
| Merid. resolution (grid points) | 790 km (8 grid points) |
| Top layer depth | 6 km |
| Bottom layer depth | 4 km |
| Typical Horizontal Scale | 1000 km |
| Typical velocity | 10 m/s |
| Coriolis parameter $F$ | $10^{-4}$ |
| Merid. gradient of $F$ ($\beta$) | $1.5 \; 10^{-11}$ |
| Orography | Gaussian hill (2 km alt.) |
| Chemical mechanism | RACM (Stockwell et al., 1997) |
| Surface pressure * | 1000 hPa |
| Temperature * | 24.9 °C |
| Boundary layer thickness * | 1.2 km |

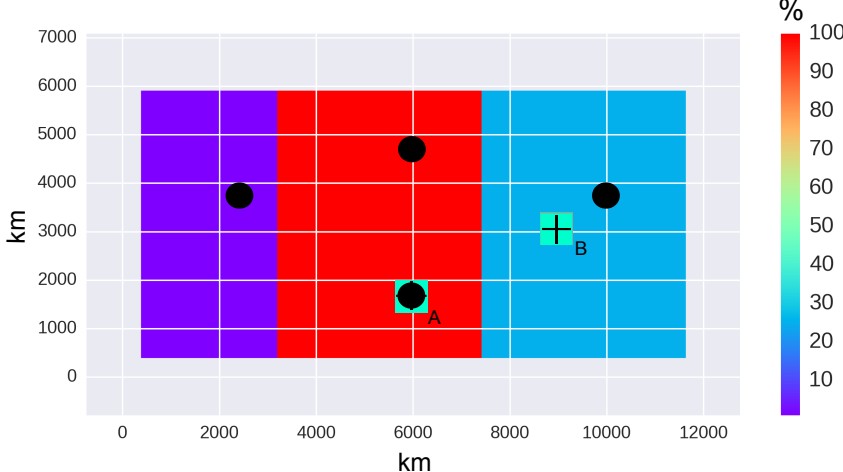

**Figure 1.** QG-Chem horizontal domain: scaling factor for the chemical surface emissions (in colors), location of the synthetic observations used in the assimilation experiments (black circles) and locations for which time series of DA experiments are displayed (crossed boxes A and B).



**Table 2.** Initial condition used to initialize the truth simulation. The initial fields are constant over the QG-Chem domain. Same values are also assigned to the meridional boundaries of QG-Chem during all simulations. Values are equal to zero for chemical species that are not listed below. Chemical concentrations are expressed in volume mixing ratio units (vmr).

| Variable | Value |
|---|---|
| Meteorology | $(m \cdot s^{-1})$ |
| $(u_1, v_1)$ | (40 , 0) |
| $(u_2, v_2)$ | (10 , 0) |
| Chemistry | (vmr) |
| $O_2$ | 0.2095 |
| $O_3$ | $30 \cdot 10^{-9}$ |
| $CO_2$ | $310 \cdot 10^{-6}$ |
| OH | $1.5 \cdot 10^{-12}$ |
| $HO_2$ | $1 \cdot 10^{-13}$ |
| $H_2O_2$ | $1 \cdot 10^{-12}$ |
| $N_2O$ | $310 \cdot 10^{-9}$ |
| NO | $0.2 \cdot 10^{-9}$ |
| $NO_2$ | $0.1 \cdot 10^{-9}$ |
| $HNO_3$ | $0.5 \cdot 10^{-9}$ |
| $HNO_4$ | $0.1 \cdot 10^{-9}$ |
| $CH_4$ | $1.6 \cdot 10^{-6}$ |
| CO | $150 \cdot 10^{-9}$ |
| Cl | $1 \cdot 10^{-12}$ |
| HOCl | $1 \cdot 10^{-13}$ |
| HCl | $1 \cdot 10^{-12}$ |
| Br | $1 \cdot 10^{-13}$ |
| BrO | $1 \cdot 10^{-13}$ |
| HBr | $1 \cdot 10^{-13}$ |

Results for four key species are considered through the study: nitrogen dioxide ($NO_2$), ozone ($O_3$), carbon monoxide (CO) and carbon dioxide ($CO_2$). The first three are of concern for air-quality, since they have an elevated toxicity and their concentration is strongly related to anthropogenic emissions. The chemical reactivity and typical tropospheric life-time of $NO_2$, $O_3$ and CO is, however, very different. $NO_2$ typically arises from the oxidation of nitric oxide (NO) in combustion processes. It is highly reactive, lasting in the atmosphere from few hours in summer to some days in winter and it is an ozone precursor. $O_3$ is a secondary gas, mainly formed by the reaction of nitrogen oxides and hydrocarbons under sunlight. In the tropopshere, it has





**Table 3.** Surface chemical emissions used for all QG-Chem experiments. Values are equal to zero for chemical species that are not listed below. Geographical scaling is further applied to these values as shown in Fig. 1.

| Species | Value ($10^9$ molec m$^{-2}$ s$^{-1}$) |
|---|---|
| NO | 121.29 |
| CO | 2500 |
| $CH_4$ | 802 |
| ETH | 6.25 |
| HC3 | 37.67 |
| HC5 | 44.43 |
| HC8 | 19.14 |
| ETE | 22.33 |
| OLT | 39.67 |
| OLI | 6.37 |
| TOL | 9.02 |
| HCHO | 5.77 |
| ALD | 14.45 |
| KET | 5.7 |
| XYL | 14.55 |
| CSL | 3.68 |

a typical atmospheric life-time of 2-3 weeks. CO is produced by partial oxidation of carbon compounds, which can occur in industrial or natural combustion processes. It also participates to $O_3$ chemistry and has an atmospheric life-time of about 1-2 months. Finally, $CO_2$ is of major concern for its effect on climate and has the longest life-time among all the considered species (>30 years). However, $CO_2$ emissions are not activated in the experiments, which makes this gas behaving like a passive tracer in our study.

Averages of model fields for the two cases are presented in Figures 2 and 3. We note in both cases the presence of zonal gradients of chemical concentrations for species that are strongly related to surface emissions (e.g. $NO_2$, $O_3$ and CO). Maps for case 2 show an increased complexity due to the developed circulation patterns. They also display the influence of meridional boundary conditions, which produce local minima in $O_3$ and CO fields in correspondence of the advection of clean air masses from outside the domain. The average model trajectory during 24 hours show significant differences among all considered species. Since all species are advected and surface fluxes are constant in time, these features arise from the complex chemical interactions and photo-chemistry. Note for example the daylight increase of $O_3$ as a consequence of $NO_2$ production from NO emissions during night-time and daytime photolysis. In contrast, CO shows an almost linear increase in time, due to costant





surface emissions and longer life-time. All the numerical experiments that are shown later in this study are done on these 2 episodes, to demonstrate the skills of DA in different atmospheric conditions.

## 3    Data assimilation algorithm

We considered two data assimilation algorithms in this study: 3D-Var and 4DEnVar. The first is the simplest type in the family

of variational DA algorithms and currently the most used in operational chemical assimilation systems (Marécal et al., 2015). It is taken as a reference against which the benefits of more complex (and costly) algorithms can be assessed. 4DEnVar is an hybrid algorithm that combines benefits of variational and ensemble methods. It is already used in a number of NWP models (Buehner et al., 2010; Lorenc et al., 2015) and was tested in the framework of meteorological toy-models (Desroziers et al., 2014; Fairbairn et al., 2013). A summary description of the two algorithms is given below, as well as some specific aspects

relative to the atmospheric chemistry implementation presented in this study. In the third section (3.3), a method based on the post-processing of 4DEnVar output is proposed to correct model biases. The Objected Oriented Prediction System OOPS (Y. Trémolet, personal communication) is a generic software framework to develop data assimilation systems and was used to implement and run all the DA experiments described in this study.

### 3.1    3D-Var

The 3D-Var analysis can be computed after a model forecast time-step by means of minimizing the quadratic cost-function $J$ (Kalnay, 2003):

$$J(\delta\mathbf{x}) = \frac{1}{2}\delta\mathbf{x}^T\mathbf{B}^{-1}\delta\mathbf{x} + \frac{1}{2}(\delta\mathbf{y}^o - \mathbf{H}\delta\mathbf{x})^T\mathbf{R}^{-1}(\delta\mathbf{y}^o - \mathbf{H}\delta\mathbf{x}) \tag{5}$$

for the increment $\delta\mathbf{x}$ used to correct the previous forecast $\mathbf{x}_b$ (also named background), i.e. $\mathbf{x} = \mathbf{x}_b + \delta\mathbf{x}$ where $\mathbf{x}$ is the control variable (e.g. the 3D chemical state). Here, $\mathbf{B}$ and $\mathbf{R}$ are the background and observation error covariance matrices, $\mathbf{H}$ the

linearized observation operator that transforms an increment of the control variable into an increment in the observation space, $\delta\mathbf{y}^o$ the difference between the observations vector $\mathbf{y}^o$ and the previous forecast $\delta\mathbf{y}^o = \mathbf{y}^o - \boldsymbol{H}(\mathbf{x}_b)$, using the non-linear observation operator. The minimization of $J$ can be achieved with standard techniques for the solution of large linear systems: the $\mathbf{B}$-preconditioned Conjugate Gradient algorithm has been used in this study (Derber and Rosati, 1989). The result of the minimization is the analysis increment $\delta\mathbf{x}_a$ (3D). To advance in time, the analysis $\mathbf{x}_b + \delta\mathbf{x}_a$ is used as the new initial condition

for the following forecast step and so forth. The relative simplicity, efficiency and robustness of the 3D-Var algorithm make it very suitable for operational models. Its practical implementation requires mainly the development of a covariance model for $\mathbf{B}$ (Weaver and Courtier, 2001). Multivariate chemical assimilation can be performed with 3D-Var by extending the 3D control variable $\mathbf{x}$ to contain multiple 3D model variables (i.e. chemical species).

In this study the control variable $\mathbf{x}$ is set to represent the complete model state, i.e. the stream function $\psi$ plus the 96

chemical species. $\mathbf{B}$ is modeled through the sequential application of 1D square-root correlation operators and multiplication





**Figure 2.** Meteorological and chemical fields for the zonal flow and clean atmosphere case (case 1). Time-averaged fields for a 24 hours period on the left, time series of domain-averaged values for the same 24 hours period on the right. The wind field and the concentration of the four chemical species of interest (CO, $NO_2$, $O_3$, $CO_2$ not shown since constant and equal to 310 ppmv) are shown from top to bottom.

**Figure 3.** Same plots as in Fig. 2 but for the Rossby waves flow and polluted atmosphere case (case 2).

by the variance:

$$\mathbf{B}\delta\mathbf{x} = \mathbf{B}^{T/2}\mathbf{B}^{1/2}\delta\mathbf{x} \qquad (6)$$



$$\mathbf{B}^{1/2}\delta\mathbf{x} = \mathbf{\Sigma}^{1/2}\mathbf{C}_z^{1/2}\mathbf{C}_x^{1/2}\mathbf{C}_y^{1/2}\mathbf{C}_v^{1/2}\delta\mathbf{x} \tag{7}$$

where $\mathbf{C}_z$, $\mathbf{C}_x$, $\mathbf{C}_y$, $\mathbf{C}_v$ are respectively the vertical, zonal, meridional, multivariate correlation operators and $\mathbf{\Sigma}$ is the variance (diagonal matrix). Except for $\mathbf{C}_x$, which is modeled in QG-Chem by means of solving a diffusion equation in spectral coordinates (Mirouze and Weaver, 2010), the other correlation operators are represented by full symmetric positive-definite matrices. The following parameters are used to set $\mathbf{B}$: one horizontal length scale that defines the decorrelation scale for the zonal and meridional coordinates, one value for the vertical correlation and one value for the chemical correlation between each couple of model variables. The variance is specified using one global value for each variable. The resulting $\mathbf{B}$ is uniform and homogeneous on the horizontal plane. More complex $\mathbf{B}$ models could be introduced, to account for example for spatial variability and heterogeneity of the background error covariance. However, this is out of the scope of the present study, which is intended to reproduce typical operational chemical DA settings, where $\mathbf{B}$ is usually specified using a single variance and correlation length for each chemical species.

Only surface observations are considered in this study. Therefore, the observation operator $\mathbf{H}$ is represented by the bi-linear interpolation of model values to the observation location. A diagonal observation error covariance $\mathbf{R}$ is used, as it is the case in most real DA systems.

One drawback of 3D-Var is that DA results rely strongly on the background error covariance $\mathbf{B}$, which should depend on the previous assimilation cycles and forecast errors (flow-dependence). In practical applications, $\mathbf{B}$ is usually set constant, estimated from verification of previous forecasts (climatological) and/or tuned to provide the best fit of the analyses against independent observations. In the case of multivariate chemical assimilation, the estimation and validity of climatological error covariances between chemical species has not yet been demonstrated. As a consequence, multivariate corrections are normally neglected ($\mathbf{C}_v = \mathbf{I}$). This simplification is also used in this study. Finally, 3D-Var provides a correction to the control variable each time the system is observed (i.e. 1 hour in air-quality applications). This does not allow to exploit the dynamical information contained in observations time series and prevent all estimation of model error terms.

## 3.2 4DEnVar

The 4DEnVar algorithm is meant to solve the main drawbacks of 3D-Var by introducing the temporal dimension in the quadratic cost function (5), similarly to what the classic 4D-Var algorithm does (Dimet and Talagrand, 1986). However, compared to the latter, it avoids the introduction of the tangent linear and adjoint codes of the forecast model. Following the notation in Desroziers et al. (2014), the 4DEnVar cost function can be written as :

$$J(\underline{\delta\mathbf{x}}) = \frac{1}{2}\underline{\delta\mathbf{x}}^T\underline{\mathbf{B}}_e^{-1}\underline{\delta\mathbf{x}} + \frac{1}{2}(\underline{\delta\mathbf{y}}^o - \underline{\mathbf{H}}\underline{\delta\mathbf{x}})^T\underline{\mathbf{R}}^{-1}(\underline{\delta\mathbf{y}}^o - \underline{\mathbf{H}}\underline{\delta\mathbf{x}}) \tag{8}$$

where all underlined terms are now time-dependent (4D). The control variable $\underline{\mathbf{x}}$ becomes the temporal trajectory of the model state ($\psi$ plus the 96 chemical species in this study). The cost function is computed for an assimilation window that can span several hours or days. The assimilation window is discretized with an arbitrary number of sub-windows, which defines the





temporal dimension of the 4D vectors and matrices in (8). A sub-window of 1 hour is used in this study, to match the surface observation frequency. The minimization of $J$ returns a 4D vector $\underline{\delta \mathbf{x}}_a$, which provides the analysis trajectory for the entire assimilation window.

The forecast error covariance $\underline{\mathbf{B}}_e$ is estimated from an ensemble of perturbed model trajectories:

$$\underline{\mathbf{B}}_e = \frac{1}{L-1}(\underline{\mathbf{x}}'_1,...,\underline{\mathbf{x}}'_L)(\underline{\mathbf{x}}'_1,...,\underline{\mathbf{x}}'_L)^T \tag{9}$$

where $\underline{\mathbf{x}}'$ are the four-dimensional perturbation and $L$ denotes the size of the ensemble. It results that $\underline{\mathbf{B}}_e$ describes spatial (3D), multivariate and temporal covariances at once. An analogy with the 4D-Var weak constraint cost function shows that $\underline{\mathbf{B}}_e$ represents a numerical approximation of the linearized forecast model and model error covariance (Desroziers et al., 2014). Note that, if the length of the assimilation window is set to one sub-window, the 4DEnVar algorithm becomes equivalent to an hybrid 3D-Var, where the forecast error covariance is specified through an ensemble ($\mathbf{B} = \mathbf{B}_e$). As for the 3D-Var case, the minimization of $J$ is achieved using a $\mathbf{B}$-preconditioned Conjugate Gradient algorithm (Algorithm n. 3 in Desroziers et al. (2014)). $\underline{\mathbf{H}}$ and $\underline{\mathbf{R}}$ are block diagonal matrices whose blocks are the operators $\mathbf{H}$ and $\mathbf{R}$ defined in Sec. 3.1 for each sub-window.

A nice property of the 4DEnVar algorithm is that the effect of the model error covariance, generally denoted by $\mathbf{Q}$ (Tremolet, 2006), can be introduced easily by adding physical perturbations during the computation of the ensemble of forecasts. This lets the model dynamics develop the complex covariances that derive from physical perturbations, without the need of specify and implement a covariance model for $\mathbf{Q}$. Whenever the sole initial condition of the ensemble is perturbed at the beginning of the assimilation window, the 4DEnVar approximates the 4D-Var algorithm, in the strong constraint formulation. The flexibility of the model error specification in 4DEnVar is a strong asset for atmospheric chemistry DA, where the sources of uncertainty can be highly variable and function of the chemical species.

The main drawback of 4DEnVar in practical applications to large scale models is related to the finite size of the ensemble. As a consequence the 4D error covariance $\underline{\mathbf{B}}_e$ is not full-ranked and covariance terms may contain statistical noise. The effect of a noisy covariance is the appearance of spurious analysis increments far away from assimilated observations, or between physically uncorrelated model variables (in the multivariate case). Since with 4DEnVar temporal covariances are also estimated through the ensemble, this effect concerns also the time dimension. This is a typical issue with all ensemble-based methods and large systems, and demands the introduction of a *localization* operator, which attenuate non-local increments. The localization is applied to $\mathbf{B}_e$ by:

$$\underline{\mathbf{B}} = \underline{\mathbf{B}}_e \circ \underline{\mathbf{C}} \tag{10}$$

where $\circ$ denotes the Schur product (entrywise product) and $\underline{\mathbf{C}}$ is a 4D correlation operator that damps non-local covariances. The numerical implementation of (10) is made under the following approximations: no time localization is applied and the same 3D (and multivariate) correlation operator $\mathbf{C}$ is used for all 4DEnVar sub-windows. It follows that $\underline{\mathbf{C}}$ is a block matrix with all elements set equal to $\mathbf{C}$. This simplification reduces significantly the numerical cost of the algorithm (Desroziers et al., 2014). Hence, for the experiments presented in this study, we could use the covariance operator described in (6) by setting the variance terms to one. The choice of parameters of the correlation operators is discussed in the results section. The





development of more accurate localization operators is an ongoing research topic (Bocquet et al., 2015) and their utilization will be considered in a future study.

### 3.3 Diagnosis of model error and forecast correction with 4DEnVar

One of the objectives of this study is to retrieve model error information from the 4DEnVar solution, which can be potentially used to improve the chemical forecasts for the next day. The 4DEnVar analysis increment $\underline{\delta \mathbf{x}}_a$ accounts for the correction of both the initial condition and the model forecast (model error). However, the control variable $\underline{\delta \mathbf{x}}_a$ only contains the model state. It follows that the correction of the initial condition is simply given by $\delta \mathbf{x}_a^{(t=0)}$, i.e. the first element of the 4D vector $\underline{\delta \mathbf{x}}_a$. The correction of the model error contributes to the values of $\delta \mathbf{x}_a^{(t>0)}$ but a diagnostic procedure has to be applied to retrieve it.

We can compute a forecast trajectory $\underline{\mathbf{x}}_f$ using the non-linear model starting from the updated initial condition $\mathbf{x}_b^{(t=0)} + \delta \mathbf{x}_a^{(t=0)}$. Subtracting it from the 4DEnVar analysis at $t > 0$, we obtain

$$\Delta \underline{\mathbf{x}}^{(t>0)} = \underline{\mathbf{x}}_a^{(t>0)} - \underline{\mathbf{x}}_f^{(t>0)}. \tag{11}$$

This difference contains information about the contribution of the model error on the 4D state, as explained below. Let us first define the model error $\underline{\boldsymbol{\eta}}$ and analysis error $\underline{\mathbf{e}}_a$ as

$$\boldsymbol{\eta}^{(t)} = \mathbf{x}_*^{(t)} - \mathcal{M}_{(t-1)\to(t)}(\mathbf{x}_*^{(t-1)}) \tag{12}$$

$$\mathbf{e}_a^{(t)} = \mathbf{x}_a^{(t)} - \mathbf{x}_*^{(t)} \tag{13}$$

where $\mathbf{x}_*^{(t)}$ is the truth at time $t$, $\mathcal{M}_{(t-1)\to(t)}$ is the integration of the model from time $(t-1)$ to time $t$, with the temporal discretization matching the length of 4DEnVar sub-windows (Sec. 3.2). From (12), we have

$$
\begin{aligned}
\boldsymbol{\eta}^{(t+1)} &= \mathbf{x}_*^{(t+1)} - \mathcal{M}_{(t)\to(t+1)}(\mathbf{x}_*^{(t)}) \\
&= \mathbf{x}_*^{(t+1)} - \mathcal{M}_{(t)\to(t+1)}(\boldsymbol{\eta}^{(t)} + \mathcal{M}_{(t-1)\to(t)}(\mathbf{x}_*^{(t-1)})) \\
&= \mathbf{x}_*^{(t+1)} - \mathcal{M}_{(t)\to(t+1)}(\boldsymbol{\eta}^{(t)} + \mathcal{M}_{(t-1)\to(t)}(\boldsymbol{\eta}^{(t-1)} + \mathcal{M}_{(t-2)\to(t-1)}(\mathbf{x}_*^{(t-2)}))) \\
&= \dots \\
&= \mathbf{x}_*^{(t+1)} - \mathcal{M}_{(0)\to(t)}(\mathbf{x}_*^{(0)}) - \sum_{j=1}^{t} \prod_{l=j+1}^{t+1} \mathbf{M}_l \boldsymbol{\eta}^{(j)}
\end{aligned}
$$

where we have used the Taylor expansion and assumed that the forecast model can be linearized inside each sub-window: $\mathcal{M}_{(t-1)\to(t)} \sim \mathbf{M}_t$. This equality can be rewritten as

$$\mathbf{x}_*^{(t+1)} - \mathcal{M}_{(0)\to(t)}(\mathbf{x}_*^{(0)}) = \boldsymbol{\eta}^{(t+1)} + \sum_{j=1}^{t} \prod_{l=j+1}^{t+1} \mathbf{M}_l \boldsymbol{\eta}^{(j)} \tag{14}$$



Using the definition (13) and the linear assumption on the model, it yields from (11) that

$$
\begin{aligned}
\Delta \mathbf{x}^{(t)} &= \mathbf{x}_a^{(t)} - \mathcal{M}_{(0)\to(t)}(\mathbf{x}_a^{(0)}) \\
&= \mathbf{x}_a^{(t)} - \mathbf{x}_*^{(t)} + \mathbf{x}_*^{(t)} - \mathcal{M}_{(0)\to(t)}(\mathbf{x}_a^{(0)} - \mathbf{x}_*^{(0)} + \mathbf{x}_*^{(0)}) \\
&= \mathbf{e}_a^{(t)} + \mathbf{x}_*^{(t)} - \mathcal{M}_{(0)\to(t)}(\mathbf{e}_a^{(0)} + \mathbf{x}_*^{(0)}) \\
&= \mathbf{e}_a^{(t)} + \mathbf{x}_*^{(t)} - \mathcal{M}_{(0)\to(t)}(\mathbf{x}_*^{(0)}) - \prod_{l=0}^{t} \mathbf{M}_l \mathbf{e}_a^{(0)}
\end{aligned}
$$

Substituting the equality (14) into the latter, we get

$$
\Delta \mathbf{x}^{(t)} = \mathbf{e}_a^{(t)} - \prod_{l=0}^{t} \mathbf{M}_l \mathbf{e}_a^{(0)} + \boldsymbol{\eta}^{(t)} + \sum_{j=1}^{t} \prod_{l=j+1}^{t} \mathbf{M}_l \boldsymbol{\eta}^{(j)}. \tag{15}
$$

Therefore, if the error $(\mathbf{e}_a^{(t)} - \prod_{l=0}^{t} \mathbf{M}_l \mathbf{e}_a^{(0)})$ is small, the vector $\Delta \mathbf{x}^{(t)}$ represents the contribution of the model error $\underline{\boldsymbol{\eta}}$ on the 4D state, accumulated through the preceding 4DEnVar sub-windows. Hereafter, we will name $\Delta \mathbf{x}^{(t)}$ *effective* model error, to be distinguished from the model error, which is generally associated with $\underline{\boldsymbol{\eta}}$.

In a first instance, $\Delta \underline{\mathbf{x}}$ can be used to diagnose the underlying presence of model error in the forecast. Furthermore, if the model error is stationary along multiple DA windows (bias), $\Delta \underline{\mathbf{x}}$ could be used to correct the forecast for the next window:

$$
\tilde{\underline{\mathbf{x}}}_f^{i+1} = \underline{\mathbf{x}}_f^{i+1} - \Delta \underline{\mathbf{x}}^i \tag{16}
$$

where the superscript denotes the assimilation window index and the $\tilde{\underline{\mathbf{x}}}_f$ is the corrected forecast. Validation of operational air-quality models shows that biases contribute for a large part of the model uncertainties (Marécal et al., 2015; Zyryanov et al., 2012; Huijnen et al., 2010), which justifies the implementation of the proposed bias correction procedure. Note that this procedure is compatible with model biases that show an hourly variability but are stationary on successive days, as it is found in most of air-quality models (Marécal et al., 2015; Gaubert et al., 2014). Note also that the computation of the effective model error is blind to the type of the underlying model error (e.g. chemical emissions or physical parametrizations). The two main requirements needed to make a useful estimation of $\Delta \underline{\mathbf{x}}$ are: i) analysis errors smaller than model errors and ii) an approximate knowledge of the sources of model error, necessary to generate informative ensembles.

Alternatively from the proposed procedure, model error terms $\boldsymbol{\eta}^{(t)}$ could be diagnosed for each sub-window and then applied to the next day forecast on hourly bases. However, this correction method is more intrusive, because it must be applied during the non-linear model forecast and was not considered for this study.

# 4 Results and discussion

Numerical experiments are described and discussed in this section. The objective is to assess the performances of the 4DEnVar algorithm for the assimilation of the four key species: $NO_2$, $O_3$, $CO$ and $CO_2$. In all experiments a model simulation with unperturbed parameters (same as in Sec. 2.3) represents the truth. The experiments are performed during the two meteorological-chemical situations described in Sec. 2.3. This permits to test DA during different chemical and meteorological regimes.





Synthetic observations are generated from the truth applying $\mathbf{H}$ (Sec. 3.1) and by adding a normally distributed error (Tab. 4). Four observation locations are considered for the experiments (Fig. 1), where chemical species are observed hourly. The relatively low density of the observation network allows to assess DA skills at unobserved locations. Model forecasts are produced by perturbing only the initial condition (perfect model), the surface emissions (model error) or both. DA is performed with either 3D-Var or 4DEnVar, and the obtained analyses are compared to the truth. The meteorology is never observed neither perturbed.

Operational air-quality centers collect hourly observations and perform DA typically once per day (Marécal et al., 2015). Therefore, a 24 hours assimilation window is adopted when using the 4DEnVar algorithm, with 1 hour sub-windows matching the observations frequency. For the same reason, 24 sequential cycles of 1 hour are adopted with 3D-Var. The main processes affecting air-quality forecasts (i.e. daily emissions, evolution of the mixing layer, photo-chemistry) have a period of approximately 24 hours. Therefore, a 24 hour window permits to account for errors on main model processes. The utilization of longer windows is theoretically possible with 4DEnVar, assuming that the linearization of the model perturbations remains valid. However, the numerical cost of the minimization increases with the windows length when keeping fixed the size of the sub-windows. The costs-benefit ratio of longer windows will deserve further investigations in real applications.

Section 4.1 presents results based on a perfect model hypothesis and compares 3D-Var and 4DEnVar performances for chemical reanalyses. Sec. 4.2 introduces model errors and is focused on the estimation of the effective model error using 4DEnVar. Sec. 4.3 summarizes the results on a larger number of DA windows: 4DEnVar results using the model bias correction procedure are compared to 3D-Var ones for both reanalyses and forecasts.

### 4.1 Perfect model experiments

Experiments considering a perfect model are presented in this section for one assimilation cycle of 24 hours. Emphasis is placed on the re-analysis capabilities of DA. A 24 hours long forecast is produced by perturbing only the initial condition. Initial perturbations are computed applying $\mathbf{B}^{1/2}$ (7) to a gaussian uncorrelated noise field $N(0,1)$ at $t = 0$. Since chemical concentrations can span different order of magnitudes in the atmosphere, the standard deviation values set in $\mathbf{B}^{1/2}$ depend on the chemical species (Tab. 4). The background error standard deviations have been chosen to be about 20-30% of the average field values (Fig. 2, 3). The same horizontal correlation length has been set for all species (750 km). Multivariate correlations in $\mathbf{B}^{1/2}$ are switched off so that perturbations of chemical species are not correlated at $t = 0$. Chemical correlations can, however, arise at $t > 0$ due to chemical couplings.

DA is applied to correct the 24 hours long perturbed forecast. The same covariance matrix $\mathbf{B}$ that was used to produce the initial perturbation is applied at each cycle of the 3D-Var analysis (1 hour). Hence, the background error covariance used in DA is perfectly known at $t = 0$, which represents the best possible case in DA. At $t > 0$ the true $\mathbf{B}$ depends on the observations assimilated at previous steps and on the model dynamics, which makes the use of a fixed $\mathbf{B}$ within 3D-Var a raw approximation. However, this is the typical setting of operational air-quality models.

With 4DEnVar, an ensemble of 16 forecasts is generated perturbing the initial condition with the same $\mathbf{B}$ as above. Therefore, the ensemble size is small compared to the dimensions of the system ($16 \cdot 8 \cdot 97 = 12416$ variables). When using small size





**Table 4.** Background ($\sigma_B$) and observation ($\sigma_O$) error standard deviation used in DA experiments (expressed in volume mixing ratio units).

| Species | Case 1 $\sigma_B$ (vmr) | Case 2 $\sigma_B$ (vmr) | $\sigma_O$ (vmr) |
|---|---|---|---|
| NO$_2$ | | $1.2 \cdot 10^{-10}$ | $0.9 \cdot 10^{-10}$ |
| O$_3$ | $6.1 \cdot 10^{-9}$ | $12.2 \cdot 10^{-9}$ | $3.2 \cdot 10^{-9}$ |
| CO | $32.4 \cdot 10^{-9}$ | $64.8 \cdot 10^{-9}$ | $8.1 \cdot 10^{-9}$ |
| CO$_2$ | $40.5 \cdot 10^{-6}$ | $40.5 \cdot 10^{-6}$ | $20.2 \cdot 10^{-6}$ |

ensembles, ensemble localization is necessary. In this study, horizontal localization is applied to the 4D ensemble covariance using Eq. (10), by setting an horizontal length scale equal to the double of values used for **B** (1500 km). Multimultivariate-variate localization is performed using a correlation coefficient of 0.5, which was chosen empirically. A sensitivity analysis concerning the ensemble size and the localization choices is presented later in this section. All vertical terms of the covariance or localization matrix are always set to zero in this study, since only the bottom layer of the QG-Chem model is considered.

### 4.1.1 Univariate assimilation

This section compares results of 3D-Var and 4DEnVar DA in univariate settings, i.e. one independent assimilation experiment is performed for each of the four chemical species. The controlled species corresponds to the species that is measured and that is perturbed at the initial time. With 3D-Var, this is obtained setting all terms of **B** to zero except for the 3D covariance of the selected species. With 4DEnVar the same is obtained setting to zero all multivariate localization coefficients.

Fig. 4 and Fig. 6 compare the temporal trajectories of the analysis of each species obtained from 3D-Var and 4DEnVar, respectively for case 1 and case 2 described in Sec. 2.3. Each figure provides the temporal trajectories at the two grid points shown in Fig. 1: one located in the polluted region and in correspondence of assimilated observations (grid point A) and one in the cleaner region and slightly displaced from the measurements location (grid point B). Note that in Fig. 4, NO$_2$ is not shown because a global variance value could not be used to perturb both the very low concentrations in the clean maritime region and the higher ones in the urbanized region.

In addition to the comparison of the two DA algorithms, these experiments also permit to assess the impact of the initial perturbation on chemical forecasts. First of all, we remark that O$_3$ forecasts are very close to the truth values after 24 hours, which is a consequence of the fact that O$_3$ is strongly controlled by precursor emissions and photo-chemistry. The memory of the initial condition is rapidly lost for O$_3$, as it was also demonstrated within regional air-quality models (Jaumouillé et al., 2012). This is not the case for CO$_2$ and CO, which have a longer life-time. Therefore, the initial perturbation is advected by the wind field and the spread between the forecast and the truth lasts longer in time. NO$_2$, which lasts few hours in a summer-time atmosphere is practically not sensitive to the perturbation of the initial condition (Fig. 6). Note also that in case 1 the chemical concentrations are always lower at the clean location (grid point B) than at the polluted one (grid point A), except for





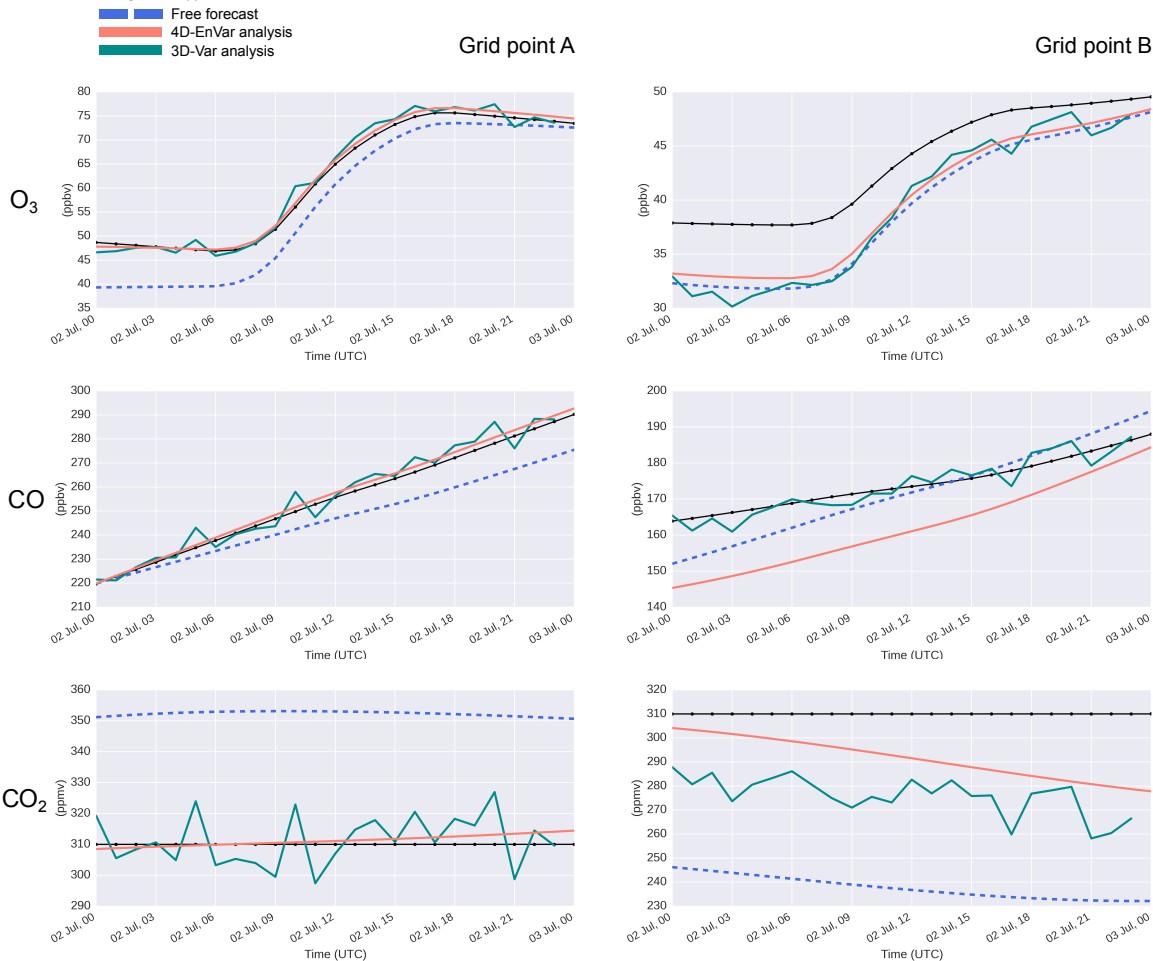

**Figure 4.** DA results for the case 1 and perfect model hypothesis. Temporal trajectories for the three assimilated species are shown from top to bottom ($O_3$, CO and $CO_2$). Forecast, 3D-Var/4DEnVar analyses and truth are shown in each plot for the grid points A (left plots) and B (right plots) depicted in Fig. 1.

$CO_2$ that is neither emitted nor chemically produced. This is a consequence of the varying emission factors (Fig. 1) and the weak transport (Fig. 5). More mixed conditions are instead present in case 2, because of the enhanced transport. The chemical concentrations at both grid points A and B can reach same values for long-lived transported species ($O_3$, CO and $CO_2$). Values remain fairly different for the short-lived $NO_2$, which stays closer to NO emissions even in presence of strong transport.

5   We remark that the 4DEnVar provide in general better analyses than 3D-Var for all species, both in case 1 and 2. First, it can be observed from Fig. 4 and Fig. 6 that the analysis time-series obtained with the 4DEnVar are smoother than those resulting from 3D-Var, because daily trajectories are optimized at once with 4DEnVar. The sequential aspect of 3D-Var, instead, makes the analysis more sensitive to the random observation errors. This introduces the observed jumps in the analyses.





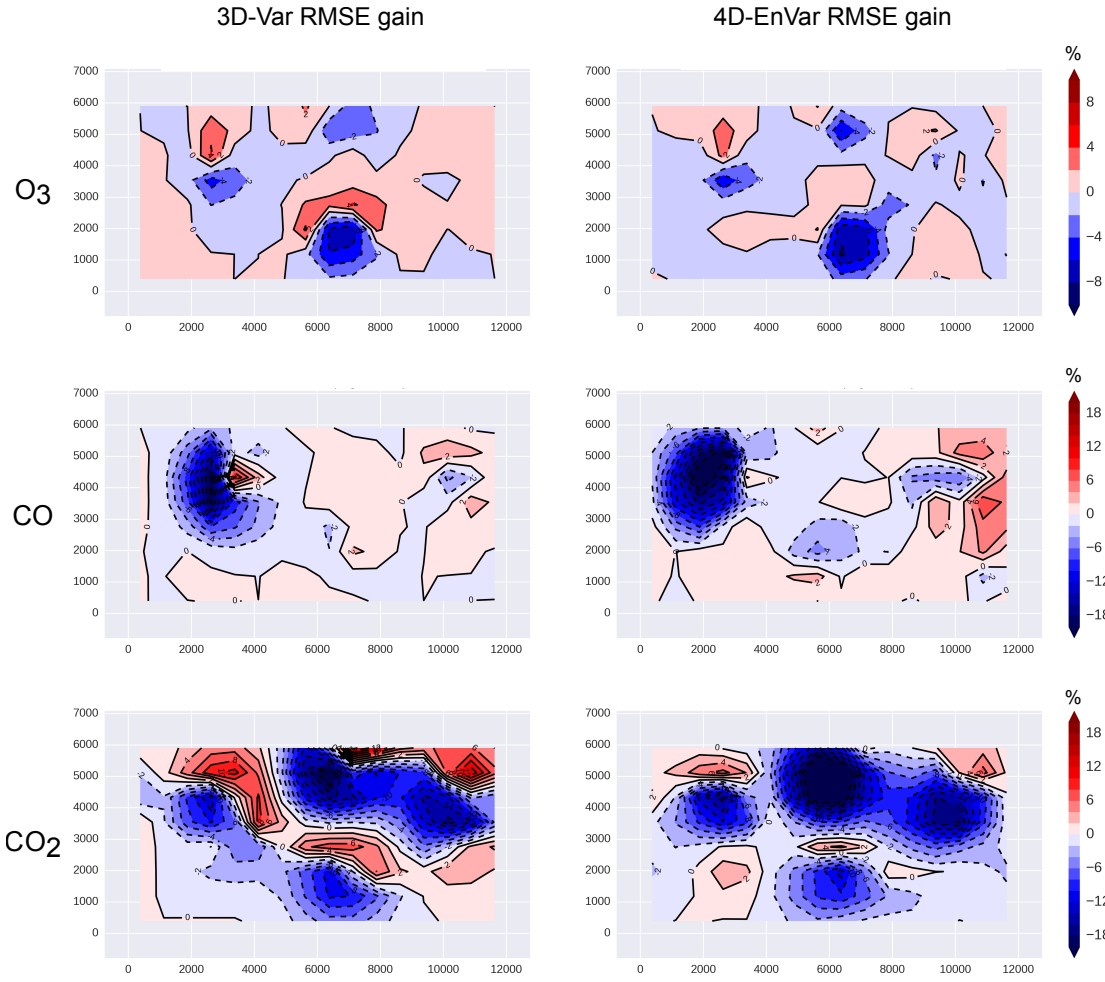

**Figure 5.** DA results for the case 1 and perfect model hypothesis. From top to bottom: the RMSE gain (17) is displayed respectively for $O_3$, CO and $CO_2$ assimilation experiments. Blue color means that DA lowered the RMSE, red color means that DA increased the RMSE. Plots on the left are obtained using 3D-Var, on the right using 4DEnVar.

Fig. 5 and Fig. 7 provide the Root Mean Square Error (RMSE) gain for every grid point $(i,j)$ of the model domain,

$$\text{RMSE}^{gain}_{(i,j)} = \frac{1}{\bar{X}_{(i,j)}}\left(\text{RMSE}^{anl}_{(i,j)} - \text{RMSE}^{fct}_{(i,j)}\right). \tag{17}$$

Here, $\bar{X}_{(i,j)}$ is the average concentration of the truth values (Fig. 2), $\text{RMSE}^{fct}$ and $\text{RMSE}^{anl}$ are the RMSE values for the forecast and analysis respectively. The RMSE at the location $(i,j)$ is defined as

$$5 \quad \text{RMSE}_{(i,j)} = \sqrt{\frac{1}{N}\left(\underline{\mathbf{x}}^*_{(i,j)} - \underline{\mathbf{x}}_{(i,j)}\right)^T\left(\underline{\mathbf{x}}^*_{(i,j)} - \underline{\mathbf{x}}_{(i,j)}\right)}, \tag{18}$$



**Figure 6.** Same as Fig. 4 but for case 2. NO$_2$ is also added among chemical species.

where $N = 24$ is the total number of sub-windows, $\underline{\mathbf{x}}^*_{(i,j)}$ and $\underline{\mathbf{x}}_{(i,j)}$ denotes the temporal trajectories of the truth and the analysis (or forecast) at the grid point $(i,j)$ respectively. In Fig. 5 and Fig. 7, blue color means that RMSE values after DA has reduced (DA improved the forecast) and red color means that RMSE values after DA has increased (DA degraded the forecast).





**Figure 7.** Same as Fig. 5 but for case 2. NO$_2$ is also added among chemical species.

Therefore, we can see from these figures that with 4DEnVar, improvements of the RMSE at unobserved locations are more pronounced than those of 3D-Var (more widespread blue regions and less red regions in RMSE gain). This is likely due to a better description of the background error covariance, which is flow-dependent and closer to the true forecast error covariance



**Table 5.** Summary statistics of RMSE gain for perfect model DA experiments in Fig. 5, 7. Signs have been inverted compared to Figures, to show positive values when DA reduces the RMSE.

| Case | Species | 3D-Var | | | 4D-EnVar | | |
|------|---------|--------------|--------------|--------------|--------------|--------------|--------------|
| | | Max. Gain (%) | Min. Gain (%) | Avg. Gain (%) | Max. Gain (%) | Min. Gain (%) | Avg. Gain (%) |
| | $O_3$ | 7.86 | −4.31 | 0.17 | 8.30 | −2.75 | 0.55 |
| | CO | 20.71 | −10.05 | 0.76 | 23.38 | −7.89 | 1.05 |
| Exp 1 | $CO_2$ | 23.55 | −16.80 | 1.72 | 27.58 | −6.90 | 3.93 |
| | $O_3$ | 11.34 | −4.73 | 0.67 | 20.83 | −4.05 | 1.77 |
| | CO | 17.23 | −3.35 | 1.22 | 18.87 | −4.14 | 1.63 |
| Exp 2 | $CO_2$ | 17.73 | −13.33 | 2.02 | 19.42 | −3.39 | 4.52 |
| | $NO_2$ | 32.49 | −6.22 | 0.17 | 30.10 | −7.33 | 0.83 |

within 4DEnVar. For example, the background error covariance used to assimilate $NO_2$ with 3D-Var is highly overestimated for $t > 0$, because the $NO_2$ field rapidly converges to the truth after few hours (Fig. 6). In this specific case, the RMSE is degraded even at observed locations (Fig. 7, bottom left plot). This is an instructive example of the effects of incorrectly specified $\mathbf{B}$ within 3D-Var. Finally, note that for other species also 3D-Var is capable of decreasing RMSE at unobserved locations. This effect is a result of the advection of the analysis increments, and, as expected, is more pronounced with long-lived tracers ($CO_2$) than with reactive or emitted gases ($O_3$, CO, $NO_2$).

Table 5 reports the minimum, maximum and average of the field values in Fig. 5 and 7, with the sign inversed to display positive values for positive gain of DA, negative otherwise. It can be seen that with 4DEnVar, the maximum degradation of RMSE (i.e. the minimum gain in absolute values) is always smaller by a factor 3 or 4 than the maximum gain, and the average RMSE gain is always positive, i.e. DA improves the forecast. This can be considered satisfactory for atmospheric chemistry, because the chemical system has a dissipative behavior and errors on the model state cannot grow during the forecast step. Since the error covariance of the initial condition is perfectly known in the experiments setup, the degradation of the 4DEnVar analysis is a consequence of the algorithm hypotheses (e.g. the linearization of the forecast model) or of the numerical implementation (e.g. the finite size of the ensemble and the localization approximations).

### 4.1.2 Multivariate assimilation

A second set of DA experiments has been performed, but perturbing and assimilating the four chemical species at the same time. Therefore, one 24 hours long forecast and the corresponding analysis has been computed for all species, instead of four independent analyses as before. In case of 3D-Var, elements of $\mathbf{B}$ related to the four assimilated species are set using the same parameters as in Sec. 4.1.1. Elements related to unobserved species are kept zero as well as cross-variable correlations. This





leads to a multi-species assimilation, which is not yet multivariate. Effects on unobserved variables or between species are still permitted by chemical couplings in the forecast model.

The gain on RMSE obtained with 3D-Var for the four species (not shown) is very similar to those obtained in Sec. 4.1.1 with independent experiments.

5     With 4DEnVar, the corresponding multi-species assimilation has been tested by setting to zero the cross-variable correlation coefficients in the localization operator. In addition, a multivariate case has also been examined, by setting the correlation coefficients to 0.5. In both cases results (not shown) were found to be again very similar to those in Fig. 5 and 7.

These results indicate that, when the sole initial condition is taken as source of uncertainty, the chemical coupling between species does not influence much DA. This is confirmed by the ensemble standard deviation of the 4DEnVar experiments in 10    Sec. 4.1.1. For each of the eight assimilation experiments the average ensemble standard deviation has been computed for all species (perturbed and not perturbed at $t = 0$). The ensemble standard deviation of unperturbed species stays below 1% of the local concentration (not shown), compared to typical values of about 10-15% for the species that is perturbed initially. Therefore, the perturbation of the sole initial state does not affect significantly the chemical balance. This also justify neglecting the cross-correlations between chemical species in operational systems that assimilate hourly observations sequentially. This 15    result was expected for weakly reacting species like $CO_2$ or CO but was not evident for reacting gases such as $O_3$ and $NO_2$.

Finally, 4DEnVar was capable to provide same good results as in Sec. 4.1.1 when enabling the cross-variable covariances (and the respective localization). This means that the noise of chemical cross-covariances due to the small ensemble size (16 members) did not degrade the results. This leaves hope for an effective multivariate chemical assimilation, when the role of chemical couplings becomes larger (Sec. 4.2).

20    **4.1.3   Ensemble size and localization**

This section examines the impact of the principal parameters of the 4DEnVar algorithm, i.e. the ensemble size (16 members in previous experiments) and the localization length scale, on the analysis RMSE. These two aspects are closely linked with the dimension of the system and the eigenspectrum of the error covariance matrices (Furrer and Bengtsson, 2007). For instance, for a strongly decaying spectrum relatively few ensemble members are enough to provide a good approximation of the error 25    covariance matrix. If this is not the case, a larger ensemble size allows in principle less severe localization (Ménétrier et al., 2015). However, no theoretical formulation exists already for the 4DEnVar that links these parameters, e.g. the localization scale with the ensemble size. Moreover, the number of parameters of the localization operator (e.g. the horizontal length scale) might depend on the chemical species. A full exploration of the parameter space becomes rapidly unpractical even in a simplified model framework. Finally, the approximations made in the implementation of the localization operator (Sec. 3.2) 30    can potentially have a larger effect than the choice of the parameters themselves. Therefore an empirical approach has been used in this paper, and we postpone a detailed theoretical analysis to future work.

In the empirical approach, first the ensemble size is examined using fixed but reasonable values for the localization operator (Sec. 4.1). The ensemble size is the main limiting factor in operational forecast centers and the smallest ensemble providing



better results than 3D-Var for all cases and species was taken. Second, the impact of horizontal and chemical localization are examined keeping the selected ensemble size fixed.

Results for the two case studies are shown in Fig. 8 (varying ensemble size), Fig. 9 (varying horizontal localization) and Fig. 10 (varying cross-variable localization). The minimum, maximum and mean RMSE gain (Table 5) of the analysis are considered to compare 4DEnVar and 3D-Var experiments. Hence, the plots display the differences between the RMSE gain of 4DEnVar minus the 3D-Var one, with the sign opportunely adjusted to display positive values when 4DEnVar beats 3D-Var, negative values otherwise. In general, the desired case is represented by all bars (mean, maximum and minimum gain) being positive. Nevertheless, a positive average gain with the sporadic occurrence of negative maximum/minimum gains can be tolerated, if the values for the minimum gain bar do not become too negative. Since the minimum RMSE gain is already negative (Table 5), negative values for the corresponding bars in Figs. 8, 9, and 10 mean that the degradation of the 4DEnVar analysis is larger than the 3D-Var one, which is not desired.

Results are very similar using 64 or 128 members, for all species and both case 1 and 2, suggesting that some convergence of the assimilation scores is achieved with more than 64 members (Fig. 8). As expected, the accuracy starts to decrease using less than 32 members, with most of the gain of 4DEnVar over 3D-Var being lost using only 8 members. The $CO_2$ case represents an exception, probably because even a very small ensemble accounts much better for the linear $CO_2$ dynamics than the static $\mathbf{B}$ of 3D-Var. In few cases, the RMSE gain occasionally decreases when increasing the size of the ensemble (e.g. the minimum gain for $O_3$, case 2). This is due to statistical fluctuations when the ensemble sizes are small. To avoid misinterpretation of statistical noise, the comparison of 3D-Var and 4DEnVar is repeated in Sec. 4.3 for a larger number of DA cycles. We remark finally, that results with 16 members satisfy the requirements expressed above: better average and maximum RMSE gain than 3D-Var for all species and cases and a limited number of cases of RMSE degradation. Hence, an ensemble size of 16 members is retained. We remind that the objective of this study is to demonstrate the applicability of a DA algorithm that outperforms currently implemented methods in operational centers, but with an acceptable compromise between computational costs and precision. Therefore, even if an ensemble size of 32 or 64 would have represented a more accurate option for this study, we found more valuable to assess the potential of 4DEnVar when computational resources might be limited.

The choice of the horizontal localization scale is more delicate, because it is intimately linked to the model dynamics, depends on the ensemble size and on the assumptions made to construct and apply $\underline{\mathbf{C}}$ (Sec. 3.2). Fig. 9 shows that, except for the $CO_2$ case, horizontal localization is necessary to obtain meaningful results with 4DEnVar. Second, increasing the localization scale to values as high as 3000 km, compared to the 750 km of the initial perturbation scale, has the effect of improving the maximum RMSE gain but also degrading significantly the minimum one. This is not desired, as explained above. The best results, considering employing the same homogeneous and global localization scale for all chemical species, can be found for the value of 1500 km.

The configuration of a multivariate localization, or chemical localization in our study, presents the same issues as the spatial one. A similar approach as in the above paragraph has been taken. The numerical experiments described in Sec. 4.1.2, which considered multivariate cases, are used to compute error differences in Fig. 10. Again, we remark that without any localization the statistical noise of the ensemble covariance degrades significantly the results compared to 3D-Var. A localization coefficient




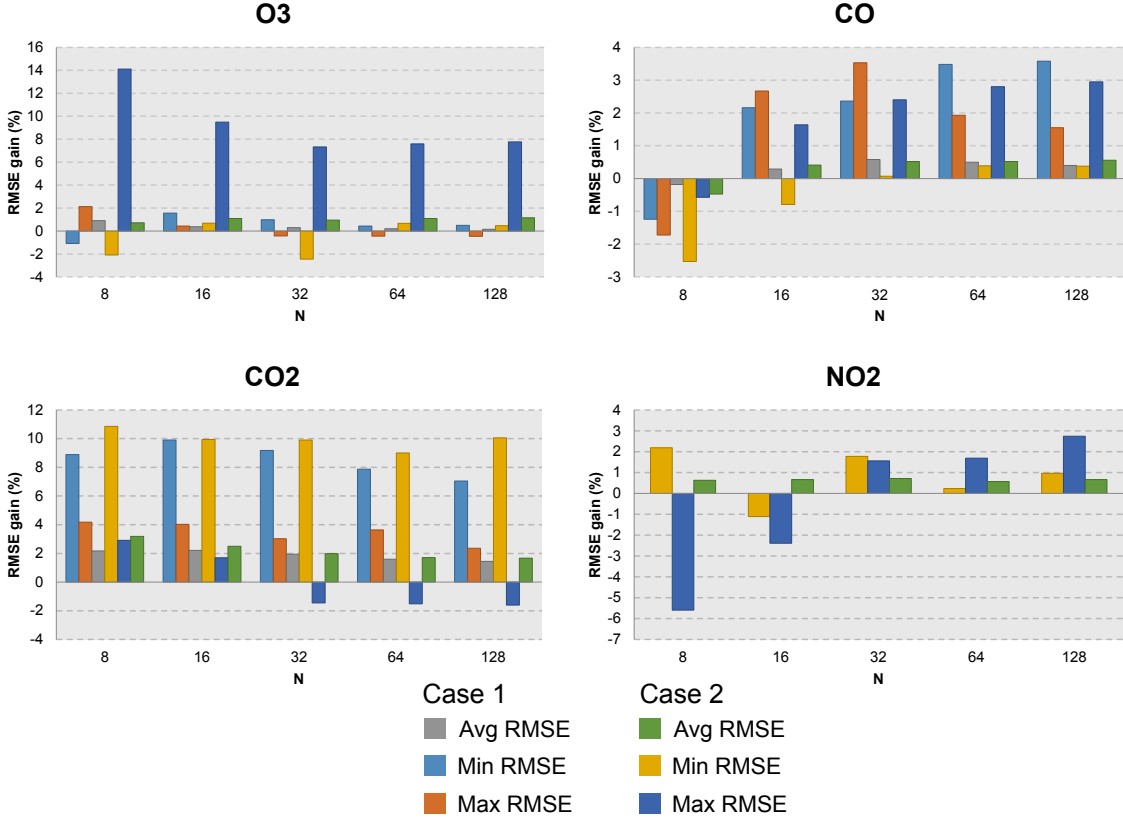

**Figure 8.** Impact of ensemble size (N) on the RMSE gain of the 4DEnVar analyses. Results are shown for the four chemical species $O_3$, CO, $CO_2$, $NO_2$. For each plot values of maximum, minimum and average RMSE gain (Table 5) are shown in different color for both case 1 and 2, compared to reference results obtained with 3D-Var. Positive values mean better RMSE gain with 4DEnVar than with the reference 3D-Var.

of 0.5 reduces efficiently the effect of noisy correlations in the 4D ensemble covariance, leaving the possibility of describing multivariate effects. We remind that multivariate effects are quite small in perfect model experiments presented until now, but can be much larger when a model error is introduced (Sec. 4.2).

We conclude that a simple localization scheme, based on global and empirically tuned parameters, provides already en-
5   couraging results for the application of 4DEnVar to large scale chemistry models. The development of more sophisticated localization operators (Bocquet, 2016) and more rigorous methods to estimate their parameters (Ménétrier et al., 2015), represents a current subject of research, and will be the topic of a future study.



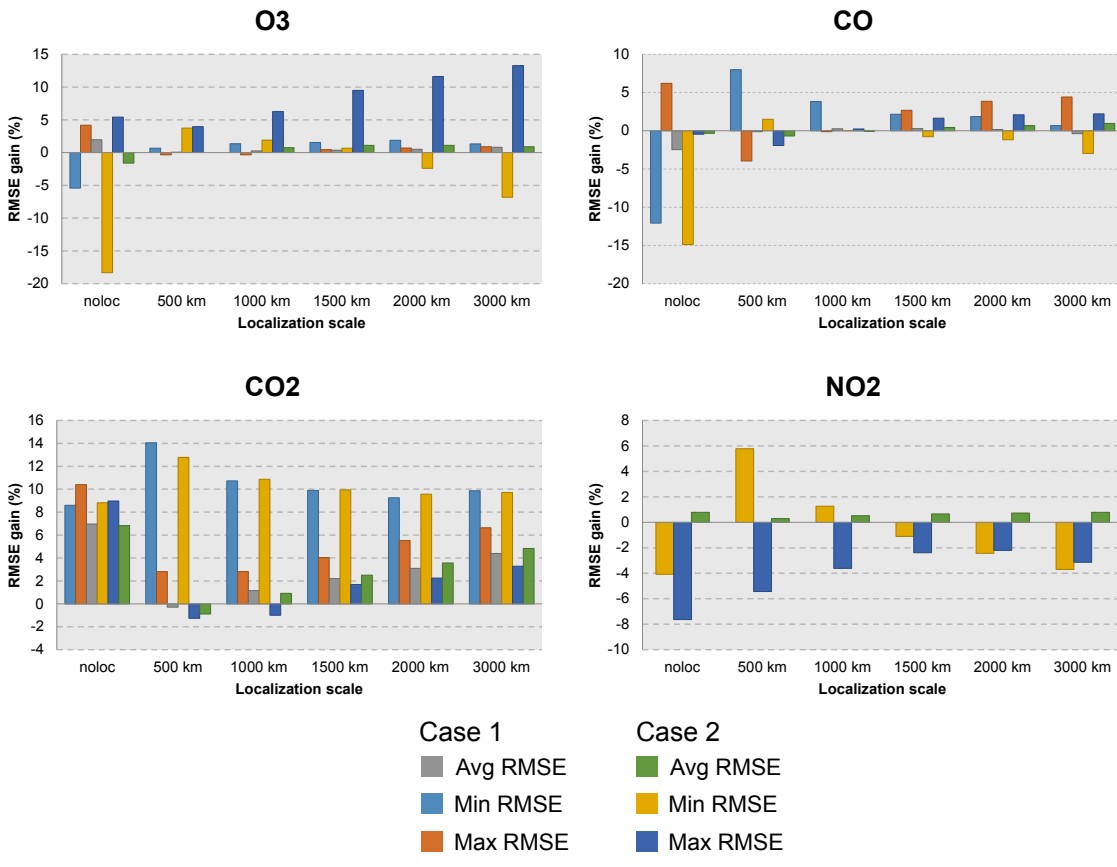

**Figure 9.** Same as Fig. 8 but for the horizontal length scale of the localization operator ('noloc' when no localization is applied).

## 4.2 Model error experiments

In the previous section we have shown how 4DEnVar is able to match or even outperform 3D-Var results when the model is perfect. However, the main interest of 4DEnVar for atmospheric chemistry arises when the model is not perfect, i.e. a case that cannot be addressed using 3D-Var or strong constraint 4D-Var. In this section 4DEnVar experiments are conducted in presence

5  of a model error term. A typical source of uncertainty in CTMs is represented by anthropogenic or biogenic emissions (Kok, 2011; Zhao et al., 2011; Ma and van Aardenne, 2004). In case of reactive species like NOx, erroneous emissions can impact strongly the formation of secondary species such as $O_3$ (Lei and Wang, 2014; Sillman, 1999). Errors on surface emissions produce already a complex and rich dynamics and will be used as a test-bed to investigate the effects of model error in DA. Other sources of uncertainties, e.g. chemistry parameters, meteorology etc. will be addressed in a future study, considering that

10  the same methodology as used here can be applied.



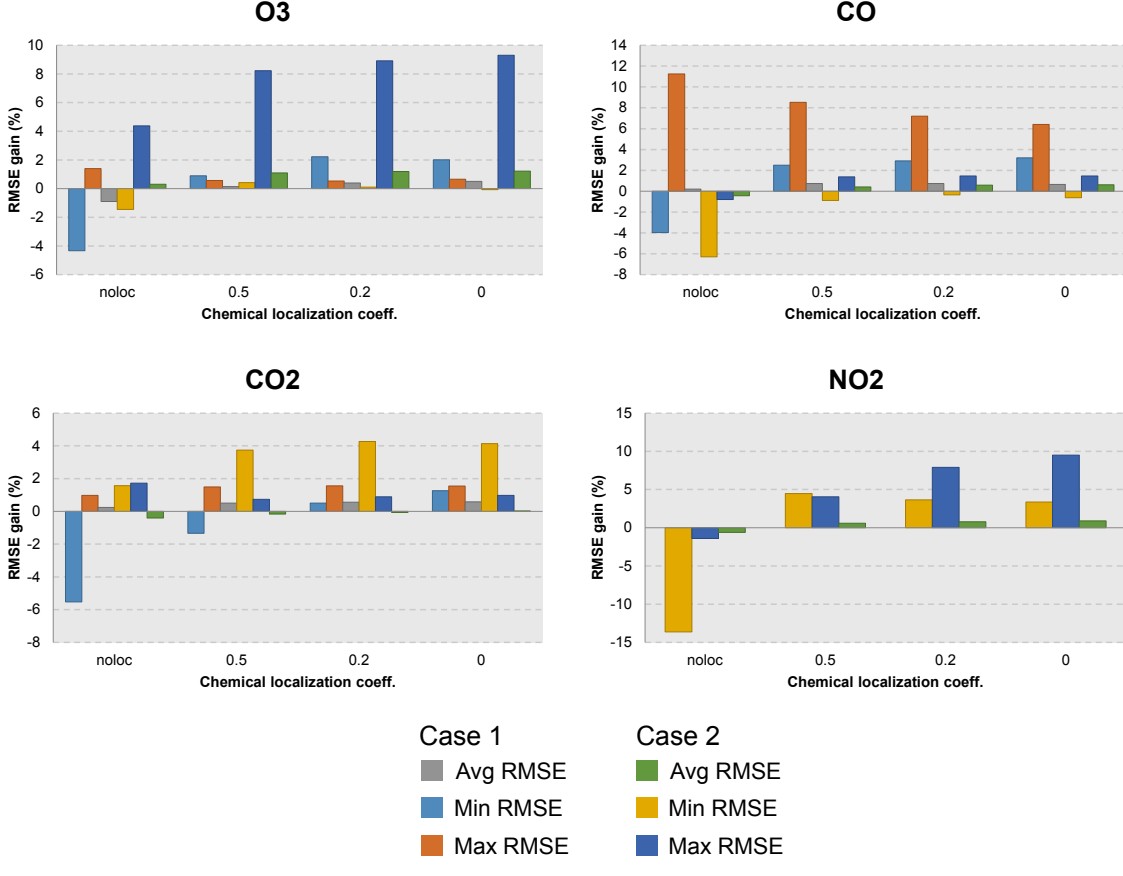

**Figure 10.** Same as Fig. 8 but for the multivariate correlation coefficient of the localization operator (0 corresponding to univariate DA, 'noloc' when no localization is applied).

Similar single cycle DA experiments are conducted as in Sec. 4.1. The truth NO emissions (Tab. 3) are perturbed by a multiplicative factor, which is sampled from a log-normal distribution with mean and sigma equal to 0.5 and 0.8 respectively. Both the forecast and the ensemble of forecasts are generated from the same distribution. Therefore, the model error covariance is assumed to be perfectly known. A multiplicative factor of 2.35 has been sampled for the NO emissions of the forecast

5  simulation. The advection has been deactivated in this set of experiments to better focus on the impact of emissions uncertainty on chemistry. A part from this, the same exact configuration as before is used for 4DEnVar (initial condition error, ensemble size, localization). Univariate $O_3$ DA is presented in Sec. 4.2.1, to analyze the effective model error estimation procedure (Sec. 3.3). Results of a multivariate DA experiment are presented in Sec. 4.2.2, to examine the combined effects of model error and chemical couplings. Finally, the impact of the model bias correction procedure is evaluated on 48 hours forecasts of several

10  species in Sec. 4.2.3.



### 4.2.1 Univariate assimilation

Three DA experiments are shown in Fig. 11: i) activating only the initial perturbation, ii) activating only the model error, iii) with both perturbations activated. Only $O_3$ observations have been assimilated and the chemical localization coefficients are set to zero to compute univariate analyses.

With the NO emissions increased by a factor of 2.35, the forecast produces higher concentrations of $O_3$ (Fig. 11, middle plot). We remark also that the effects of model error on $O_3$ appear later in the day, due to the photo-chemistry, whereas the perturbation of the initial condition has a larger effect in the first part of the day. In this specific case it is also interesting to note that, when both perturbation are applied, a compensation of the two errors cancels out the differences between the forecast and the truth later in the day. This is an example of compensating errors in atmospheric chemistry, which might be hard to detect

when comparing model results to observations.

   The 4DEnVar analyses agree well with the truth in all 3 cases, similarly to what was obtained in Sec. 4.1.1. This satisfies the main requirement for a meaningful computation of the effective model error (Sec. 3.3). The estimation of the effective model error with Eq. (11) is also displayed in Fig. 11. We remind that the effective model error is expressed in the same physical units of the model state, i.e. chemical units (ppbv). The *true* effective model error is computed by subtracting the truth from a

forecast initialized from the truth, and it is also added to the figures.

   The effective model error is zero in the first experiment, which is correctly diagnosed by the procedure. In the second experiment, the estimated effective model error is approximately zero until 10 am and grows to positive values of about 8 ppbv later in the afternoon, which is coherent with the underlying perturbations and chemical mechanism. We note that the effective model error estimation is very accurate in the second case, whereas a small bias is found for the third case between 12 and 15

UTC. Larger bias for the third case can be explained based on the increased number of degrees of freedom for the errors (initial condition plus model errors) compared to the second case. Since the ensemble size and the number of assimilated observations are kept fixed, DA becomes more difficult on the third case. We remind that in the latter, $O_3$ trajectories are also affected by the above-mentioned error compensation in the evening. If, for example, observations of $O_3$ were available only in the late afternoon, no discrimination between the two sources of error could have been achieved. Note that the bias on the effective

model error corresponds to a bias in the 4DEnVar analysis, coherently with Eq. (15).

   The temporal average of the effective model error (Fig. 12) shows that the larger errors are diagnosed in the center of the domain, coherently with the characteristics of the emissions errors (global scaling of NO emissions). The geographical patterns and the differences from the true effective error (right plot in Fig. 12) are a result of the observations location and localization scale. With the proposed procedure, the effective model error estimation relies on the localization scale that has been used

for the 4DEnVar analysis. Therefore, the effective model error approaches zero moving far from assimilated observations, no matter which the spatial patterns of the true model error are. In the third experiment, the temporal behavior of the effective model error is well captured at the observed location. However, significant differences with the second experiment are visible in the spatial distribution of the error. Ideally, the estimation of the effective model error should provide exactly the same results in the second and third experiment. Differences arise because DA is not perfect due to: small ensemble size, linearization





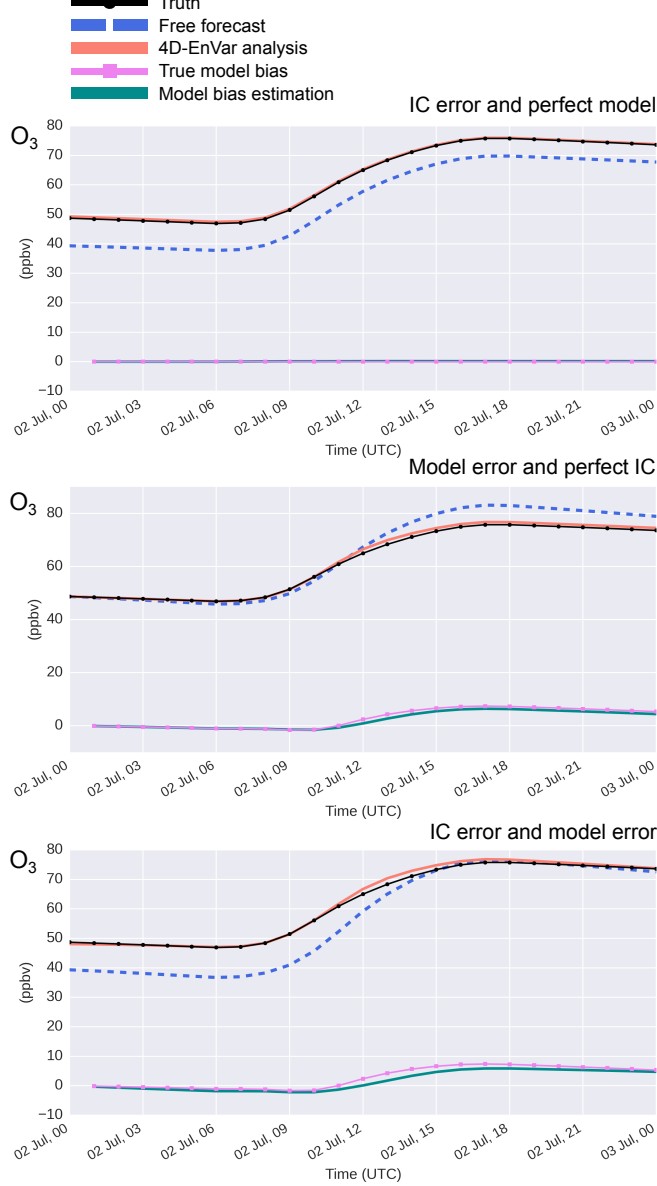

**Figure 11.** $O_3$ effective model error estimation in the univariate case (only $O_3$ assimilated). Temporal trajectories of forecast, analysis, truth, true effective model error and effective model error estimated using Eq. (11), for the pixel A located close to an observation. From top to bottom the following experiments are shown with the uncertainties being introduced: i) only in the $O_3$ initial condition (as in Sec. 4.1) ii) only in the forecast model (through surface emissions perturbation) iii) both in the $O_3$ initial condition and in the forecast model.

hypotheses, observations number and observations errors. Also, when adding degrees of freedom to the sources of uncertainty and keeping constant the observation network, DA becomes more challenging, as explained previously. However, thanks to the





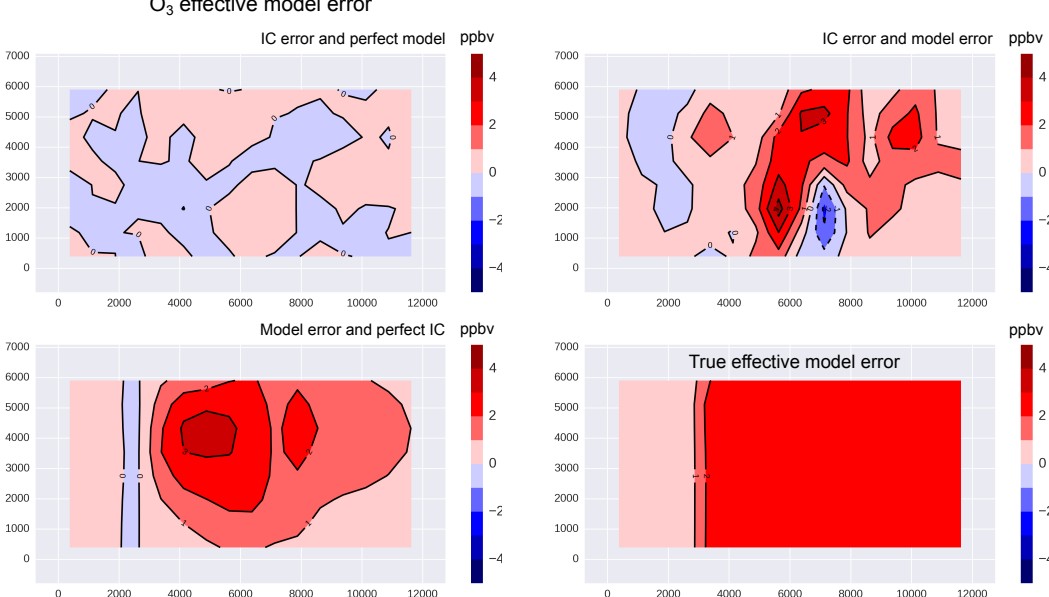

**Figure 12.** Temporal averages (24 hours) of the estimated effective model error for the three experiments as in Fig. 11. From left to right and from top to bottom, experiments i, ii and iii are shown. On the bottom-right plot the true effective model error in the experiments ii and iii is shown (no model error was activated for the first experiment).

hourly frequency of $O_3$ observations, the temporal and large scale features of the model error have been retrieved also in the third case.

### 4.2.2 Multivariate assimilation

The effects of $O_3$ assimilation on other chemical species are shown in Fig. 13. These were obtained by repeating the third
experiment of Sec. 4.2.1 but setting the chemical localization coefficient to $0.5$. Compared to Sec. 4.1.2, multivariate corrections are now very significant. For example, analyzed NO and $NO_2$ concentrations are almost halved and the initial forecast errors greatly reduced. On the other hand, CO, which was not perturbed initially nor it is strongly coupled to NO/$NO_2$/$O_3$ concentrations, is not modified at all by the DA. This shows that multivariate aspects of chemical DA can be well captured by the 4DEnVar algorithm, even with a small ensemble.

The effective model error can be computed for all the variables of the state vector, and is plotted in Fig. 13 and 14. The error fields of $NO_2$ and NO reproduce well the temporal and spatial features of the perturbation on NO emissions that was implemented in the forecast. Note that since NO is rapidly converted into $NO_2$ during night, the initial linear increase of the effective model error is only observable on $NO_2$. On the other hand, during daytime, a strong presence of the effective model error is found for both NO and $NO_2$. However, the $NO_2$ and NO errors trajectories are not linear during daytime, due to the
complex photo-chemistry.





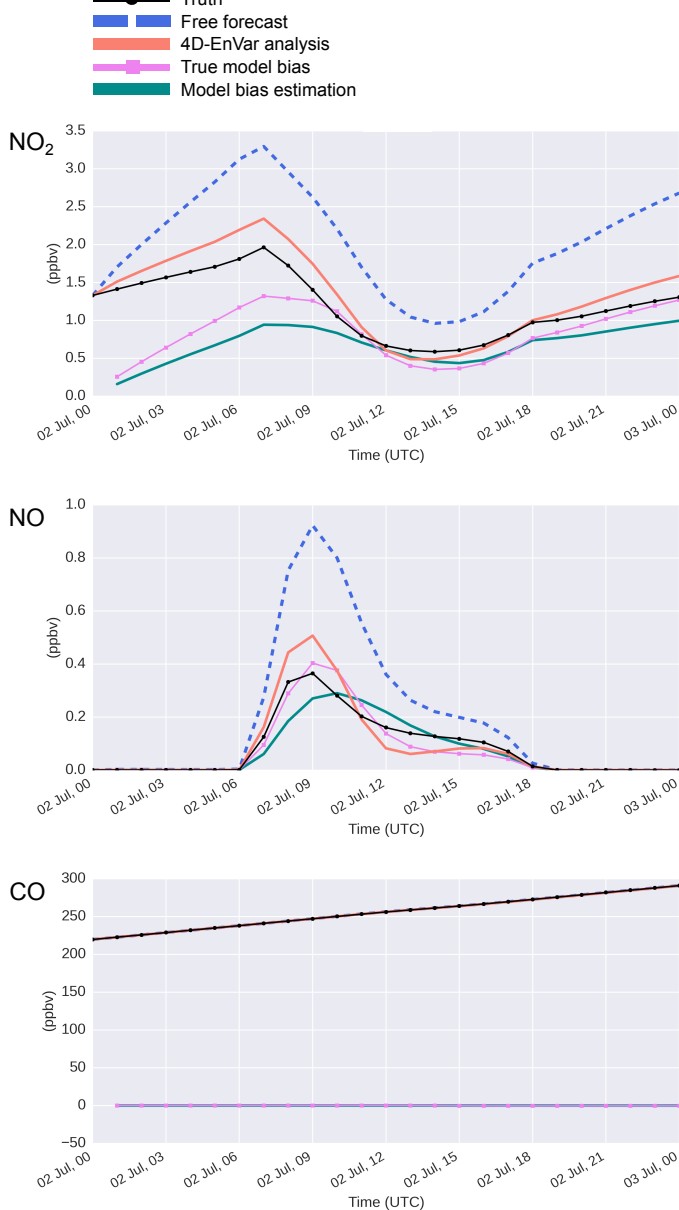

**Figure 13.** Multivariate effective model error estimation. Same plots as in Fig. 11 but for $NO_2$, NO, CO species (not assimilated), obtained assimilating only $O_3$ but with a non-zero multivariate localization coefficient. Uncertainties were both in the $O_3$ initial condition and in the forecast model (experiment iii in Fig. 11).

Strong non-linearities of the chemical system cannot be taken into account by the current implementation of the 4DEn-Var algorithm. When the $NO/O_3$ relationship were strongly non-linear, inaccurate analyses and, therefore, estimations of the effective model error have been found (not shown). This difficulty could be overcome by introducing external loops within





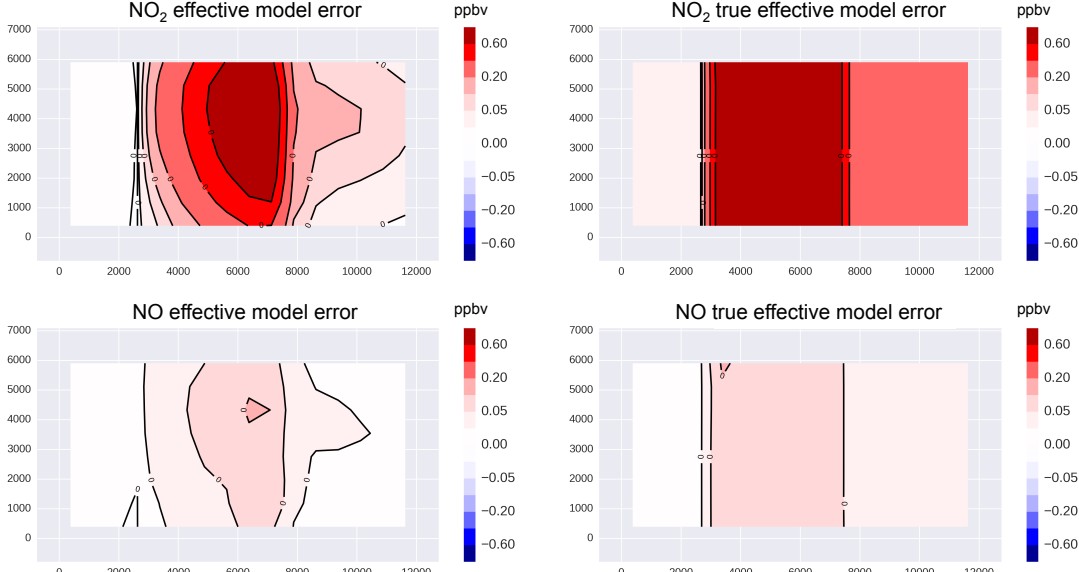

**Figure 14.** Temporal averages (24 hours) of the estimated effective model error for the multivariate DA experiment in Fig. 13. Estimated effective model error for $NO_2$ on top and NO on bottom. On the right the true effective model error is displayed for both species.

4DEnVar. This represent the objective of a future study. Alternatively, the combined assimilation of $O_3$ and $NO_2$ or a more severe chemical localization can reduce the occurrence of analysis errors in non-linear regimes.

### 4.2.3 Impact on chemical forecasts

The analysis computed in the previous section has been used to initialize chemical forecasts for the following 48 hours. This

was done to evaluate the potential of the forecast bias correction procedure (16). The corrected forecasts (CF) of $O_3$, $NO_2$ and NO are compared to the initial 3 days long forecast (OF), the 48 hours forecast initialized from the latest available analysis without any correction (AF) and the truth (Fig. 15).

First, the AF converges very rapidly (in about 12 hours) to the OF for all species, confirming the limited effects of the state correction on the chemical forecasts for the next days (Wu et al., 2008). On the other hand, the CF is very close to the

10 truth during the first 24 hours, indicating that the hypothesis of a stationary effective model error on successive days (but hourly variable) seems appropriate in this case. During the third day, a positive forecast correction is still achieved. However, chemical concentrations have evolved too much to be efficiently corrected using the bias estimated on the first day. Estimating and correcting the forecast tendencies, instead of the state, could provide a better result for the third day.

Accounting for model errors, either bias or tendencies, for the next day forecast requires the model uncertainties to be

stationary. A stationary error has been used in this study since surface emissions were taken constant in time. Most of regional air-quality models seem affected in large part by stationary errors (Marécal et al., 2015) and assuming the persistence of the bias during 24 hours looks reasonable. However, the main sources of uncertainties of real CTMs need first to be identified to



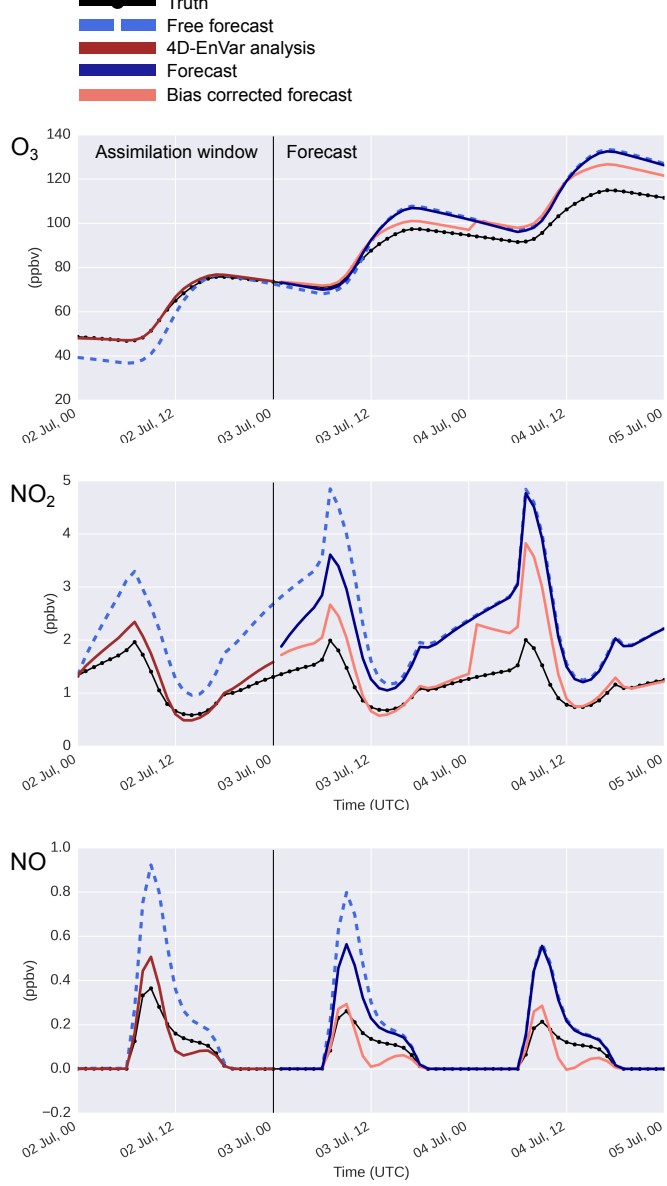

**Figure 15.** Impact of model bias correction on 48 hours forecasts. Temporal trajectory of free forecast (3 days), analysis (24 hours), truth and 48 hours long forecasts initialized from the latest available analysis, for the pixel A in Fig. 1. DA of $O_3$ is performed only during the first 24 hours (as in Fig. 13). Curves displayed in dark blue are when no bias correction is applied to the final forecast, in salmon when the effective model error estimation in Fig. 13 is used to correct the final forecast (16).

allow a meaningful effective model error estimation. Therefore, the implementation of 4DEnVar to a real CTM is necessary to further demonstrate its potential for air-quality forecasts.



**Table 6.** DA RMSEs for 14 cycles of analyses/next day forecasts (seven cycles of 24 hours starting from case 1 and 2 respectively) where all considered species are perturbed and assimilated and with model error enabled. $\epsilon_{min}$, $\epsilon_{max}$, $\epsilon_{avg}$ are the minimum, maximum and average values of the relative RMSEs (18) on the QG-Chem domain.

| Case | Species | Re-analysis | | | | | | Forecast | | | | | |
| | | 3D-Var | | | 4D-EnVar | | | 3D-Var | | | 4D-EnVar | | |
| | | $\epsilon_{min}$ | $\epsilon_{max}$ | $\epsilon_{avg}$ | $\epsilon_{min}$ | $\epsilon_{max}$ | $\epsilon_{avg}$ | $\epsilon_{min}$ | $\epsilon_{max}$ | $\epsilon_{avg}$ | $\epsilon_{min}$ | $\epsilon_{max}$ | $\epsilon_{avg}$ |
|---|---|---|---|---|---|---|---|---|---|---|---|---|---|
| | $O_3$ | 1.9 | 18.0 | 8.8 | 1.2 | 16.8 | 7.6 | 5.0 | 30.0 | 15.2 | 3.7 | 25.7 | 11.2 |
| | CO | 1.5 | 75.9 | 30.3 | 0.8 | 60.0 | 17.9 | 13.3 | 163.5 | 73.6 | 6.3 | 121.2 | 36.9 |
| Exp 1 | $NO_2$ | 11.0 | 249.2 | 48.0 | 5.6 | 63.4 | 22.8 | 29.0 | 154.8 | 85.3 | 15.9 | 134.7 | 43.7 |
| | $CO_2$ | 3.1 | 19.1 | 10.1 | 1.4 | 18.6 | 9.8 | 4.1 | 17.1 | 8.3 | 4.4 | 17.2 | 8.8 |
| | $O_3$ | 1.6 | 13.6 | 8.0 | 1.2 | 13.0 | 7.3 | 2.9 | 13.9 | 7.5 | 2.6 | 14.3 | 6.4 |
| | CO | 1.3 | 43.1 | 17.2 | 1.1 | 28.0 | 13.2 | 6.7 | 91.2 | 37.5 | 7.4 | 51.3 | 22.2 |
| Exp 2 | $NO_2$ | 8.8 | 81.5 | 37.2 | 4.2 | 64.4 | 21.5 | 19.2 | 115.7 | 69.2 | 16.0 | 80.2 | 36.6 |
| | $CO_2$ | 3.1 | 15.3 | 8.6 | 1.4 | 17.2 | 8.6 | 1.7 | 13.7 | 5.7 | 1.9 | 13.4 | 7.0 |

### 4.3 Validation on multiple DA cycles

Single windows DA experiments have been examined so far, to better analyze new aspects of 4DEnVar DA for atmospheric chemistry. In this section, DA experiments are conducted for multiple consecutive days, to provide a statistically robust comparison between 3D-Var and 4DEnVar. A general case including initial condition and model errors for all four species is considered, and both reanalyses and 24 hours forecasts for the next day are evaluated. Same values as before (multivariate experiments in Sec. 4.1.2) are used for the initial perturbation and for the algorithm settings. Compared to Sec. 4.2, all emissions in Tab. 3 are now perturbed, using a log-normally distributed scaling factor.

Synthetic observations are generated from a 30 days long truth simulation (as in previous sections). The four species are assimilated for two periods of seven days, starting respectively 2nd July (case 1) and 20th July (case 2). However, at the end of each cycle of 24 hours (forecast, analysis and next day forecast), the initial condition is reinitialized with the truth concentrations. Initial condition perturbations are recomputed advancing the seed of the pseudo-random generator. Therefore, the 14 daily cycles are statistically independent. This is done to increase the experiment statistics, without having to deal with the propagation of the analysis covariance through consecutive cycles. This aspect will be investigated in a future study.

The relative RMSE between the experiments and the truth is computed at each grid point for the two seven days long periods (case 1 and 2) as in Eq. (18). Results are summarized in Tab. 6: minimum, maximum and average values of the relative RMSEs computed over the QG-Chem domain are reported for 3D-Var and 4DEnVar experiments.

We confirm precedent findings concerning the reanalyses capabilities of 4DEnVar, which provides superior results to 3D-Var for all species and in both weak and strong advection cases. Results are particularly good for species strongly related to





emissions (CO and $NO_2$), which show halved RMSE values compared to 3D-Var results. This is a consequence of precisely accounting for emissions uncertainties within 4DEnVar. The maximum RMSE is also always lower in 4DEnVar reanalyses (except for the $CO_2$ in case 2), which suggests that the occurrence of degraded results compared to 3D-Var (negative bars in Fig. 8) is not systematic.

Similar conclusions can be drawn for the RMSE of the next day forecast. We remind that the bias correction procedure has been used with 4DEnVar, whereas no correction is applied with 3D-Var. As expected, forecast RMSEs are higher than reanalyses ones for both 3D-Var and 4DEnVar. However, CO and $NO_2$ forecasts, show significantly lower RMSE with 4DEnVar, due to the forecast bias correction. $O_3$ forecast improvements are larger during case 1 than 2. This can be explained based on the differences of the chemical regimes between case 1 and 2, $O_3$ being less sensitive to NO emissions in case 2 (polluted

atmosphere). With 4DEnVar, forecasts of $CO_2$ are slightly worse than with 3D-Var, in both cases. Since $CO_2$ is not affected by the considered model error (emissions) and it is not chemically coupled to other species, the bias correction term should be strictly equal to zero for $CO_2$. However, the small ensemble size and use of localization introduce statistical noise in the effective model error estimation, which can impact the forecast correction. This issue can be mitigated by selectively setting to zero the chemical localization coefficient between species that are chemically uncoupled. However, results remained on pair

with 3D-Var in this study.

We can conclude that the forecast of species related to surface emissions, either directly or through chemical couplings, can be significantly improved when the model error is considered within DA. The forecasts of $CO_2$, which is a passive tracer in our study, seem instead dominated by the number of assimilated observation, no matter which DA algorithm is employed. However, this is not the case in real applications, where $CO_2$ concentration is modulated by uncertain anthropogenic emissions

and natural sinks.

## 5    Conclusions

The objectives of this study were i) to develop a new toy-model framework to test advanced data assimilation (DA) algorithms for atmospheric chemistry ii) demonstrate the potential of the 4DEnVar method for air-quality or more in general tropospheric chemistry applications. In particular, we addressed the questions of: how to jointly assimilate chemical species with very

different life-times and possible chemical couplings, and how to account for model error to improve forecasts for the next day.

An atmospheric chemistry reduced-order model (QG-Chem) has been developed, based on quasi-geostrophic meteorology and a detailed tropospheric chemistry scheme. It has been used to simulate the complex spatio-temporal patterns of reactive gases ($NO_x$,$O_3$) and long-lived species (CO,$CO_2$) under the effects of emissions, chemistry and transport. QG-Chem has been coupled to a generic library for data assimilation (OOPS) and has been proven to be well suited to perform a large number of

DA experiments. Concerning temporal aspects and assimilated observations, the experiments have been designed based on the implementation of chemical DA in operational air-quality forecast centers.

A number of DA experiments have been conducted to compare 4DEnVar analyses with 3D-Var ones in a perfect model hypothesis, in different meteorological and chemical regimes, in an univariate and a multivariate setting. The sensitivity of



4DEnVar results to the ensemble size and localization method was also assessed. Results with 4DEnVar are generally better for all chemical species even using a small ensemble size of 16 members, provided that ensemble localization, even if basic, is applied. This suggests that considering the linearized model dynamics to derive a flow dependent background error covariance can be beneficial for chemical reanalyses. Thanks to 4DEnVar, this can be obtained without need of tangent linear and adjoint

codes of the complex CTM. Multivariate effects were found to be not significant when a perfect model is used, suggesting that multivariate DA goes together with model error for atmospheric chemistry applications.

The justification of using an ensemble method for operational air-quality DA, which is significantly more costly than 3D-Var, has been demonstrated when model errors were introduced. It has been shown that 4DEnVar is able to take in account heterogeneous errors on chemical emissions and complex chemical couplings to cross-correct precursor species (e.g. NO and

$NO_2$), based only on hourly observations of secondary species ($O_3$). Using long enough DA windows (24 hours in this study), 4DEnVar allowed to account for the different time-scales of the chemical mechanism and the corresponding effects in DA. For example, the delayed impact of NO emissions errors on afternoon $O_3$ concentrations was correctly accounted by 4DEnVar. The contribution of model errors on the 4D chemical state can be estimated at the cost of an additional forecast, providing quantitative information on possible forecast biases, and, in case of stationary errors, a method to correct next day forecasts.

This has been tested with success on multiple cycles of DA, showing that 24 hours forecasts of $NO_2$ and CO can be twice as accurate with 4DEnVar than with 3D-Var. Significantly better results have been also found for $O_3$ forecasts in the most reactive regime.

We conclude that 4DEnVar provides a practical and powerful algorithm for chemical DA. Among the main benefits we reckon: the implicit specification of a flow-dependent and multivariate error covariance matrix, the possibility of accounting

for model errors through stochastic perturbation of model parameters, and the opportunity of using DA windows long enough to catch typical features of model errors in air-quality models. The computational cost of 4DEnVar is higher than 3D-Var, but a small ensemble of at least 15 to 20 members remain affordable within most operational centers.

The application to a real CTM remains necessary to evaluate if the advantages of 4DEnVar shown in this study hold in real applications, where model uncertainties are not perfectly known, as well as observational ones. In addition, research

on algorithmic aspects of 4DEnVar is needed to implement more accurate localization operators, to account for non-linear chemical regimes and to correctly propagate the analysis covariance through successive DA windows. QG-Chem will represent a useful tool for this type of studies.

**Code availability**

The QG-Chem code is copyright of the CERFACS laboratory. The sources and the data used in this study are available upon

request to E. Emili (emili@cerfacs.fr) or D. Cariolle (cariolle@cerfacs.fr).





*Acknowledgements.* We acknowledge Y. Tremolet and M. Fisher for providing the OOPS DA library and the QG model sources. We thank Philippe Moinat for the ASIS model sources and technical help. We finally thank Gérald Desroziers for the useful discussions on the 4DEnVar algorithm. This work was supported by the HERMES project, funded by the French LEFE INSU program.



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
