# Peer review of "Accounting for model error in air-quality forecasts: an application of 4DEnVar to the assimilation of atmospheric composition using QG-Chem 1.0"

_Geoscientific Model Development, 2016_

## Referee Comment (RC1) · Anonymous Referee #1 · 24 Jul 2016

**1   Main or general comments**

This is a very interesting paper in at least two respects: use of a 2D low-model (as opposed to the 1D model of Haussaire and Bocquet (2016) named L95-GRS) and the introduction of an offline debiasing technique. Yet, the paper suffers from partially im-balanced assumptions: while it should get a bit more realistic than the L95-GRS model, QG-Chem is sometimes less. It also occasionally suffers from too strong and unsup-ported claims that should be mitigated. Moreover, it could also easily be shortened a bit.

Here are my main recommendations, followed by more detailed ones.

1. Some of the assumptions made for the model are rather imbalanced leading to very unrealistic conditions. In the one hand, you choose a detailed gaseous chemistry, which is fine. But on the other hand you neglect deposition which is very unrealistic to obtain an interesting dynamical equilibrium of the chemical species (typical of air pollution in the boundary layer), but which is not difficult to model.

2. At some point, advection is neglected. This assumption is really too strong since it leads to a collection of 0/1-D box/column-models, whereas the purpose of this paper is more on the 4D-EnVar aspects where advection is critical.

3. The claim of not localizing in time is actually partially wrong; that is only an approximate statement. The dynamics of a consistent space-localization operator within the time window of 4D-EnVar was explored in Bocquet (2016), from which it can be understood that in the absence of time-localization the localization operator satisfies a Liouville equation.

4. You are occulting the fact that the IEnKS has a fully nonlinear variational analysis which you don't have (this remark also applies to the current implementations of 4D-EnVar in meteorology). This is important because this could prove an asset when dealing with highly non-linear air quality models.

5. Actually 4D-EnVar should be compared to the EnKF. This is pretty obvious *for assimilators* using both ensemble filter and variational methods. Your standpoint looks like a biased one from specialists only used to variational methods. I do not encourage you to redo everything replacing 3D-Var with the EnKF. But several of your statements should be mitigated.

[Figure]

6. From my understanding, the bias removal technique that you propose is actually a parametrized one since you pick up the perturbations that you apply to the ensemble. This is just a stochastic variant of the parameter estimation technique. If I am correct, you should substantially mitigate your claim on this point.

7. To shorten the paper, I would suggestion to get rid of the first configuration. It is unrealistic and I believe a bit detrimental to the paper anyway.

8. Please number all your equations. Avoiding excessive numbering is usually reserve for books, not for articles especially meant for peer-reviews.

I would recommend minor revisions. Yet, they are substantial in numbers and several of them, if justified, are mandatory in my opinion.

**2   Minor points or comments related to the main points**

1. Page 1, line 2-3, "using a reduced-order chemical transport model based on quasi-geostrophic dynamics": this statement is ambiguous since we do not understand whether it is a CCMM or a CTM. Please clarify.

2. Page 1, line 4: "to a generic software library for data assimilation": which one? This meant to become a GMD paper. That type of info should be mentioned even in the abstract.

3. Page 1, line 12: "analysys" $\longrightarrow$ "analysis".

4. Page 1, line 12-13: "A comparison with results of 3D-Var, widely used in operational centers, shows that, for some species, analysys and next day forecast errors can be halved when model error is taken in account": A similar result was obtained by Haussaire and Bocquet (2016). They showed that by using an

ensemble forecast of the meteorology, thus partially accounting for model error, the root-mean-square error of IEnKS (a nonlinear 4D-EnVar method) on the low-order online tracer model L95-T was improved by 25% to 50%.

5. Pages 3, line 6: "next day..." ⟶ "Next day...".

6. Page 3, line 13-14: Haussaire and Bocquet (2016) also did a similar experiment simultaneously estimating gaseous concentrations and fluxes using a nonlinear 4D-EnVar.

7. Page 3, line 15: "Results seems promising but still relies on the assumption that the model is perfect, i.e. that there are no additional sources of uncertainties in the model forecast other than the controlled variables (i.e. the initial state and the selected emissions)": This is a rather biased statement. The model *per see* is not assumed perfect anymore, but only part of it, such as the dynamical part. This should be explained in a less biased way. If you estimate several parameters of the parametrization of a CTM, you are morally assuming that the model is imperfect. It is just that model error is parametrized.

8. Page 4, line 4-5: "The IEnKS (Bocquet and Sakov, 2014) is a fixed-lag ensemble Kalman smoother formulated under perfect model assumptions, which can also be used to estimate erroneous model parameters through an augmented state formalism (Haussaire and Bocquet, 2016)." No! the IEnKS is an *iterative* ensemble Kalman smoother. This is quite different from the standard "fixed-lag ensemble Kalman smoother"! It is better described as a nonlinear 4D-EnVar method. Please correct. For instance: "The iterative ensemble Kalman smoother (IEnKS, Bocquet and Sakov, 2014) is a nonlinear 4D-EnVar formulated under perfect model assumptions, which can also be used to estimate erroneous model parameters through an augmented state formalism (Haussaire and Bocquet, 2016)."

   Moreover, the augmented state formalism applied to the IEnKS was first demonstrated in Bocquet and Sakov (2013).

9. Page 4, line 8-10: "These type of approaches are generally referred in the literature as ensemble-variational EnVar (Lorenc, 2013), as opposed to "hybrid" methods, which make use of ensembles only to specify error covariances matrices in variational algorithms (Belo Pereira and Berre, 2006)."

   - Instead of "These types of approaches..." it would be much better to write "These approaches", to avoid any ambiguity in the rightful claim that the IEnKS, the 4D-Var-EnKS and the 4D-EnVar are ensemble variational techniques.

   - What you call the "hybrid" methods are in fact called EDA standing for *ensemble of data assimilation* methods.

10. Page 4, line 18: "the need in IEnKS to select a number of model parameters among all the possible erroneous parameters in complex CTMs": yes, but the EnKS and the IEnKS can account for stochastic perturbations in the integration step of the ensemble as already demonstrated in Haussaire and Bocquet (2016), section 3.2, configuration Offline 2. Please amend your statement.

11. Page 4, line 23-25: "To the knowledge of the authors, EnVar type methods have not yet been implemented in air-quality or atmospheric chemistry models and no previous study has already examined the potential of 4DEnVar for chemical DA": Without any ambiguity, they have been tested in Haussaire and Bocquet (2016). I do not see any problem in recognizing that fact. Please correct.

12. Page 4, line 26: "Then" ⟶ "Than".

13. Page 5, line 6: "This allow" ⟶ "This allows".

14. Page 5, line 14: "under a generic library for data assimilation": Again, which one? Please be specific.

15. Page 6, line 18-19: "For all the experiments presented in this study a coarse resolution of 16x8 grid points has been used.": That is quite a low resolution and

in contradiction with one of the early promises: "This choice permits to examine the behavior of DA in presence of complex gradients of wind fields and vorticity". Please revise or tune down your promises.

16. Page 6, line 21-22: "The only desired property is to obtain wind fields that exhibit typical patterns of the complex atmospheric circulation." which you don't have with such a resolution. I am not criticizing your choice but the claims that are not matched by what you present. Please rephrase.

17. Page 7, line 5: "(Cariolle D, personal communication)": Daniel Cariolle is the third author. Remove this or give more details.

18. Page 7, line 6: "which has a special treatment of the Jacobian matrix". This is too vague. Please be more specific.

19. Page 7, line 13: "The meridional boundary conditions for chemical species are set to climatological values.": Ok, but what type of numerical boundary conditions? That's important.

20. Page 7, line 14-15: "Moreover, no physical removal process for the chemical species has been included in the model so far.": This is both problematic (because this a key process of air pollution modeling) and odd (because it is not so really difficult to implement). Actually accounting for removal processes here is as important as having a fine, realistic meteorology. This is quite a weakness of your paper. The budget of all species is strongly affected. This also leads to an unbalanced photochemistry (induced by a wrong ratio of precursors).

21. Page 7, line 21: How does this configuration relate to regional air pollution modeling? Please elaborate.

22. Page 8, Fig. 1: Did you show the grid? If not, could you please do so. That would help.

23. Page 10, table 3: Please give the extended name of each species in a another column.

24. Page 12, Figure 2: Haussaire and Bocquet (2016) have more realistic values of ozone for regional air pollution with a simplified CCMM than with your model! Please discuss this. Besides, the absence of a clear daily cycle for ozone is worrying and cast doubts.

25. Page 13, Figure 3: The magnitude and variation of ozone concentration is not realistic. I would have thought it should for such a toy-model.

26. Page 13, "by the variance:": You mean by the covariance matrix?

27. Page 14, lines 3-9: What is the point in using the diffusion equation trick to obtain $\mathbf{C}_x$ for such toy-model and such 1D-correlation function? And why not for $\mathbf{C}_y$?

28. Page 14, lines 3-9: Why did you use a 2D correlation function $\mathbf{C}_{x,y}$? There, using the diffusion equation would have been more meaningful(?).

29. Page 14, line 25: "The 4DEnVar algorithm is meant to solve the main drawbacks of 3D-Var": Of course that is not its primary purpose. Please replace "is meant to" with "can".

30. Page 7, line 28: "written as :" $\longrightarrow$ "written as:"

31. Page 14, 31-32: "The cost function is computed for an assimilation window that can span several hours or days": How long is the data assimilation window in your study? That is a key value that must be mentioned and discussed, including in the paper at this point.

32. Page 15, line 10: "an hybrid 3D-Var": What is an hybrid 3D-Var for you? What you have described does not look like what is usually understood as an hybrid algorithm.

33. Page 15, line 29: "...no time localization is applied and the same 3D (and mul-tivariate) correlation operator C is used for all 4DEnVar sub-windows": As I ex-plained, this is not really a "no localization" condition. Please mention this.

34. Page 15, line 32: "Hence, for the experiments presented in this study, we could use the covariance operator described in (6) by setting the variance terms to one.": The sentence is confusing. I would write "Hence, in order to specify $C$, we could use the covariance operator described in (6) by setting the variance terms to one."

35. Page 16, line 1: "is an ongoing research topic (Bocquet et al., 2015)" should be "is an ongoing research topic (Bocquet, 2016)"

36. Page 16, lines 19-23: There is an approximation here. These are not strict in-equalities. You should use at least one $\simeq$ symbol.

37. Page 16, line 26: Again, you should use the $\simeq$ symbol here.

38. Page 17, before section 4: How are the perturbations generated? This is a critical part of the EnVar schemes, rigorously derived in the IEnKS (Bocquet and Sakov, 2014), and approximately so in other EnVar systems (so far).

39. Page 18, line 1: "and by adding a normally distributed error": What did you do with the negative measurements?

40. Page 18, line 5-6: "The meteorology is never observed neither perturbed.": so this is a CTM-like experiment. This should be emphasized since this is critical.

41. Page 18, line 22: "gaussian" $\longrightarrow$ "Gaussian".

42. Page 18, line 22: $1$ in the Gaussian distribution should actually be the identity matrix.

43. Page 19, line 1: "ensemble localization" ⟶ "localization of covariances".

44. Page 19, Table 4: That would be good to have the relative value of the stds, *i.e.* divided by a standard concentration value.

45. Page 19, line 5: "All vertical terms of the covariance or localization matrix are always set to zero in this study, since only the bottom layer of the QG-Chem model is considered": I do not understand the justification. Could you please clarify?

46. Page 19, line 11: "Fig. 4 " ⟶ "Figure 4".

47. Page 19, line 20-21: "The memory of the initial condition is rapidly lost for O3, as it was also demonstrated within regional air-quality models (Jaumouillé et al., 2012).": I doubt Jaumouillé et al. (2012) were the first to show/discuss this. Please give an earlier reference in addition to yours unless I am mistaken.

48. Page 20, line 4: "...which stays...": what is "which" referring to? Please clarify.

49. Page 21, line 1: "Fig. 5 a,d Fig. 7" ⟶ "Figures 4 and 7".

50. Page 21, line 5: This definition is a bit confusion since Eq.(17) has a normalization and Eq.(18) has not. Please clarify or use non-confusing notations.

51. Page 24, line 10: "satisfactory" is inappropriate. Be more specific. *Per se* the RMSE do not mean much because of what you wrote. Only the comparison of the RMSEs between the 3D-Var and 4D-EnVar is relevant. This comparison yields satisfactory results.

52. Page 25, line 25: "If this is not the case, a larger ensemble size allows in principle less severe localization (Ménétrier et al., 2015).": The reference to Ménétrier et al., 2015 is inappropriate here (fully justified later on). This is very well known in

ensemble data assimilation (especially for the EnKF) for 15 years. Cite an EnKF
paper instead, or nothing since this is common knowledge.

53. Page 25, line 33: "The ensemble size is the main limiting factor in operational
forecast centers...": This is not always true. For instance in Meteorology Environ-
ment and Climate Change Canada is using large ensembles, an option that they
prefer (to better deal with inflation and localization).

54. Page 26, line 16: "In few cases..." ⟶ "In a few cases...".

55. Page 26, line 21-22: "We remind that the objective of this study is to demonstrate
the applicability of a DA algorithm that outperforms currently implemented meth-
ods in operational centers..." A biased statement since other ensemble-based
methods are ignored.

56. Page 27, Fig. 8: Unfortunately the average RMSE of case 2, which I consider as
the most enlightening indicator in this figure cannot be seen very easily because
of the larger bars of the maximum. I suggest that you multiply by 5 the average
values or, alternatively, provide a second y-axis on the right.

57. Page 28, line 4: "...i.e. a case that cannot be addressed using 3D-Var or strong
constraint 4D-Var." Indeed, but you could address it with an EnKF, and it has
already been. Please mitigate your statement.

58. Page 29, line 4: "A multiplicative factor of 2.35 has been sampled for the NO
emissions of the forecast simulation...": I do not understand "has been sampled";
please clarify.

59. Page 29, line 5-6: "The advection has been deactivated in this set of experiments
to better focus on the impact of emissions uncertainty on chemistry.": This seems
too strong an assumption to me! the model becoming 1D on the vertical. Some of

your conclusions are based on these experiments which have a limited scope because of this assumption. Advection is of course critical for qualitatively realistic atmospheric chemistry modeling.

60. Page 32, line 12: "during night" ⟶ "during the night".

61. Page 33, Fig. 13: the useful subtitles of each panel, as seen in Fig. 11, have disappeared. Please add them.

62. Page 33, line 3: "This difficulty could be overcome by introducing external loops within 4DEnVar": this has been partly addressed in Haussaire and Bocquet (2016). Please mention it.

63. Page 36, line 1: The title of section 4.3 "Validation on multiple DA cycles" is very misleading, since you are not cycling the scheme, but only repeating one-cycle experiment, i.e. gathering statistics. Please rephrase.

64. Page 38, line 6: "The justification of using an ensemble method for operational air-quality DA, which is significantly more costly than 3DVar, has been demonstrated when model errors were introduced": This has been demonstrated long before by many teams using RRSQRT, EnKF for air quality. Please rephrase.

65. Page 38, line 7-16: Many of the statements there should be mitigated: systematically recall that these results have been obtained in the context of a simplified model (with unrealistic features). Generalized statements for 4D-EnVar cannot be made.

66. Page 38, line 15: "...on multiple cycles of DA...": this is very misleading, as discussed before! Strictly speaking this is wrong. Please remove this statement.

67. Page 38, line 18: "We conclude that 4DEnVar provides a practical and powerful algorithm for chemical DA.": Again, you have to mitigate this statement. "In the context of a low-order/simplified model, we conclude". Also powerful is too much.

[Figure]

68. Page 38, line 26: "QG-Chem will represent a useful tool for this type of studies."
⟶ "QG-Chem could represent a useful tool for this type of studies."

**References**

Bocquet, M.: Localization and the iterative ensemble Kalman smoother, Q. J. R. Meteorol. Soc., 142, 1075–1089, doi:10.1002/qj.2711, 2016.

Bocquet, M. and Sakov, P.: Joint state and parameter estimation with an iterative ensemble Kalman smoother, Nonlin. Processes Geophys., 20, 803–818, doi:10.5194/npg-20-803-2013, 2013.

Bocquet, M. and Sakov, P.: An iterative ensemble Kalman smoother, Q. J. R. Meteorol. Soc., 140, 1521–1535, doi:10.1002/qj.2236, 2014.

Haussaire, J.-M. and Bocquet, M.: A low-order coupled chemistry meteorology model for testing online and offline data assimilation schemes: L95-GRS (v1.0), Geosci. Model Dev., 9, 393–412, doi:10.5194/gmd-9-393-2016, 2016.

Jaumouillé, E., Massart, S., Piacentini, A., Cariolle, D., and Peuch, V.-H.: Impact of a time-dependent background error covariance matrix on air quality analysis, Geosci. Model Dev., 5, 1075–1090, 2012.

---

## Referee Comment (RC2) · Anonymous Referee #2 · 8 Aug 2016

Review of paper: Accounting for model error in air-quality forecasts: an application of 4DEnVar to the assimilation of atmospheric composition using QG-Chem 1.0

By E Emili, S. Gurol, and D. Cariolle

General

This paper explores the idea of using the 4DEnVar approach for chemical data assimilation (DA).  That method is currently widely used for NWP, but as explained, chemical DA stills relies on simpler approaches such as 3DVar or objective analysis.  This is done in the context of a much reduced (toy) system defined by two layers and a limited domain.  However a comprehensive chemistry model is used. The paper is well written and demonstrates the author's excellent understanding of key issues on the current state of chemical DA.  Results are well presented, showing the added value expected from 4DEnVar as well the possibility to account for model errors.  I recommend publication after addressing the minor points below.

Minor points

It seems that are tests are done setting to zero inter-species error correlation.  While some justification is provided for this, perhaps an actual figure would help. For example on p. 23 it is said: "Finally, 4DEnVar was capable to provide same good results as in Sec 4.1.1 when enabling the cross-variables covariances".  Results are not shown to back this statement?

It is said that "Only surface observations are considered in this study" (p13, L28).  There is a need to acknowledge as well the lack of vertical propagation of information from these surface variables in the context of the "toy" system.   This represents a significant challenge in a real system.

Typos:

P3 line 6: "Next day", not "next day)

P5 L6" "This allows", not "allow"

P18 L 18, correct typo "Multimultivariate-variate" ?

P 19 L 16: "lifetime", not "life-time"

P23 L3: "this also justifies", not "justify"

P28 L 6 " Apart", not " A part"

---

## Editor Comment (EC1) · S. Remy (Editor) · 29 Aug 2016

**General comments**

This paper describes a newly developed toy model GQ-Chem for testing different data assimilation algorithms in the context of air quality forecasting. In addition, the authors present a series of cases testing existing widely used 3D-var methods against a relatively new technique, untested in an air quality framework, called 4DEnvar. The authors demonstrate the utility of GQ-Chem and the potential of the new data assimilation algorithm to address various key problems associated with forecasting air quality with chemistry transport models.

The paper clearly fits within the scope of GMD as the authors present a new model and test a new technique in the context of air quality and chemistry transport models. The paper is overall very well written and clear, but with numerous minor grammatical errors, and a couple of sections of unclear text. The methods and results are well presented. Scientifically, this paper is very interesting in the context of operational air quality forecasting, as the authors appear to demonstrate a method that addresses a set of key problems associated with forecasting and data assimilation for air quality without sacrificing, too much, model complexity or computational cost.

I therefore recommend this paper for publication following minor corrections outlined in the comments below.

**Specific comments**

I found the last paragraph in section 2.1 to be unclear and somewhat confusing. For instance, it is not clear what was meant by "The sources of the QG model provided by ECMWF…". Do the authors mean that the code for the dynamical model was given to them by ECMWF? There are some further examples listed below. I therefore recommend that the authors check this text carefully for its clarity and re-write it. Examples:

"The boundary conditions are taken cyclic…", but it is not clear what "cyclic" means in this context.

"For all the experiments presented in this study a coarse resolution of 16x8 grid points has been used." I have assumed that this is the horizontal resolution, but it is not clear whether 16 refers to the North-South dimension or the East-West. The authors should also write the horizontal resolution in terms of spatial resolution in km.

Figure 1 would probably be improved with an explanation of what the grid lines represent. One presumes these are the outlines of the grid boxes used in the model, but the authors should state this clearly.

Page 24, lines 1-2. The authors should probably remind readers that only the initial conditions are perturbed during these tests, and therefore NO2 converges to the truth due to the shorter lifetime of NO2 combined with the fact that there are no errors in the emissions.

Page 25, line 15. Can the authors think of any specific reasons why ozone and NO2 show similar behaviour to CO and CO2?

The explanations for the emission perturbations at the beginning of Section 4.2 could benefit from slightly more explanation. It isn't stated directly, but I assume from the existing text that the perturbations result in reductions in NO emissions. This should be stated directly. Also, does the size of the perturbations vary with time, and if so at what frequency?

Please can the authors add some further text to the Figure 12 caption to explain what the red and blue colours represent.

**Technical comments**

Page 1, line 5. I would recommend not using "reckon" in this context. I suggest changing from "Among the assets of 4DEnVar, we reckon the possibility to deal with multivariate aspects of atmospheric chemistry and to account for model errors of generic type." to "The assets of 4DEnVar include the possibility to deal with multivariate aspects of atmospheric chemistry and to account for model errors of generic type."

Page 1, line 10. Recommend removing comma "…of surface chemical emissions, for two meteorological and chemical…".

Page 1, line 12. Misspelling of analysis "…for some species, analysys and next day forecast errors…".

Page 1, line 22. Change to "ash" singular. "…modeling of volcanic ashes…".

Page 2, line 19. Change "…involved into rapid chemical reactions…" to "…involved in rapid chemical reactions…".

Page 2, line 31. Change "However, the model dynamics is neglected…" to "However, the model dynamics are neglected…".

Page 3, line 4. Recommend changing "More important,…" to "More importantly,…".

Page 3, line 6. Error in the text due to repetition and a fullstop that should not be there: "next day model forecast (Wu et al., 2008). next day forecasts of reactive gases such as O3 or NO2,".

Page 3, line 7. Suggest changing "…,depend weakly on the initial condition…" to "… that depend weakly on the initial conditions…".

Page 3, line 19. Recommend "EnKF naturally includes model uncertainties…" in place of existing text.

Page 3, line 25. Recommend using "determine" in place of "reckon" as determine sounds less informal.

Page 3, line 28. Life-times should not be hyphenated and should all be one word.

Page 4, line 12. Recommend changing "…which still lack for most of the operational CTMs…" to "…which are still lacking for most of the operational CTMs…".

Page 4, line 19. "…supports naturally multivariate…" to "…naturally supports multivariate…".

Page 5, line 11. "We remind that the objective…" to "We remind readers that the objective…".

Page 5, line 14. Remove comma "…assimilation, to ease the exchange…".

Page 5, line 28. Add "the" such that it now reads "…DA in the presence of complex…".

Page 5, line 29. Again missing "the" prior to "…presence of…".

Page 6, line 2. It should really be "2-Layer QG model". The plural is not needed.

Page 6, line 19. "We remind that…" add "readers" before "remind".

Page 7, line 17. "A summary of the chemical configuration is given in Table 1." Do the authors mean Table 2?

Page 7, line 20. Recommend changing to "…and a relatively clean atmosphere".

Page 7, line 21. Recommend changing to "…and a polluted atmosphere".

Page 7, line 4. Recommend changing to "…this gas behave like…".

Page 10, line 2. Consider changing to "It also participates in O3 chemistry…".

Page 10, line 13. Spelling error of 'constant' "…due to costant…". I also recommend adding 'a' prior to 'constant'.

Page 11, line 2. Remove comma "…episodes, to demonstrate…".

Page 15, line 19. Recommend changing to "and a function of the chemical species.".

Page 17, line 23. Do the authors mean to say "forecast on an hourly basis.".

Page 18, line 5. Please change to "The meteorology is never observed nor…".

Page 18, line 21. There is no need to hyphenate "reanalysis".

Page 18, line 23. Please change to "different orders of magnitudes".

Page 19, line 2. Please change to "a horizontal length". It should be 'a' instead of 'an' in this case.

Page 19, line 2. I think there is a typo here when you say "Multimultivariatevariate". Please check.
Page 19, line 11. Figure 6 seems to be introduced before Figure 5 is mentioned.

Page 19, line 22. Please change to "…which lasts a few hours in a summer…".

Page 20, Figure 4. The top left panel seems to have been cut off by accident.

Page 20, line 3. Please change to "…can reach the same values…".

Page 20, line 4. Please change to "…emissions even in the presence…".

Page 20, line 5. Please change to "…4DEnVar provides…".

Page 20, line 7. Remove comma "…from 3D-Var, because daily trajectories…".

Page 22, line 3. Add comma before 'respectively' like so: "analysis (or forecast) at the grid point (i, j), respectively.".

Page 24, line 2. Please change to "…the truth after a few hours…".

Page 24, line 18. Please change to "In the case of 3D-Var,…".

Page 24, line 19. Please change to "…species are kept at zero as well as for cross-variable correlations.".

Page 25, line 8. Recommend changing "…when the sole initial condition is taken as source of uncertainty,…" to "…when the initial condition is solely taken as a source of uncertainty…".

Page 25, line 12. Please change to "…for the species that are perturbed initially.".

Page 25, line 13. Please change to "This also justifies neglecting…".

Page 25, line 16. Please change to "…was able to provide similarly good results as in Sec. 4.1.1…".

Page 26, line 16. Please modify text "In a few cases,…".

Page 26, line 21. Recommend modifying text to "We readers remind that the…"

Page 26, line 24. Please change to "…found it more valuable…".

Page 28, line 4. Please change to "…experiments are conducted in the presence…".

Page 28, line 6. Please change to "In the case of reactive species…".

Page 29, line 1. Recommend changing to "The true NO emissions…".

Page 30, line 13. I recommend changing the text to "We remind readers that…". I recommend the same change for page 30, line 22.

Page 30, line 22. Recommend changing to "…more difficult for the third case.".

Page 30, line 31. Recommend changing to "…matter what the spatial patterns…".

Page 31, line 1. Please modify text to "…observation number and observation errors.".

The labelling of the colour bar for Figure 12 has been cut off. Please correct this.

Page 32, line 7. Please modify text to "…nor is it strongly coupled to…".

Page 33, line 2. Please modify text to "When the NO/O3 relationship was strongly non-linear,…".

Page 34, line 1. Please modify text to "This represents the objective of…".

Page 34, line 5. I think the reference to Eq. 16 needs to be modified to include Eq. Check for other instances.

Page 36, line 5. Please change text to "The same values as before".

Page 36, line 17. Please correct text to "We confirm preceding findings".

Page 37, line 5. Please correct text to "We remind readers that the bias correction…".

Page 38, lines 18-19. Recommend changing text from "Among the main benefits we reckon:…" to "We determine the main benefits to be:…". Reckon sounds informal in this instance.

Page 38, line 27. Change to "…for these types of studies."

All instances of "i.e." and "e.g."should be written as: ", i.e.," and ", e.g.,". Such that both are bracketed at the front and back by commas.

When using 'respectively', the authors should pay attention that it is always placed in parenthetical commas if in the middle of the sentence or just before it if used at the end of a sentence, i.e., "…, respectively,…" or "…, respectively.".

---

## Author Comment (AC1) · 15 Sep 2016

**Accounting for model error in air-quality forecasts: an application of 4DEnVar to the assimilation of atmospheric composition using QG-Chem 1.0 . Reply to referee # 1**

Emanuele Emili[1], Selime Gürol[1], and Daniel Cariolle[1]

[1]CECI UMR5318 CNRS/CERFACS, Toulouse, France

*Correspondence to:* Emili (emili@cerfacs.fr)

**1   Reply to general comments**

We thank the reviewer for his comments, which helped to improve the manuscript. The detailed replies follow:

1. *Some of the assumptions made for the model are rather imbalanced leading to very unrealistic conditions. In the one hand, you choose a detailed gaseous chemistry, which is fine. But on the other hand you neglect deposition which is very unrealistic to obtain an interesting dynamical equilibrium of the chemical species (typical of air pollution in the boundary layer), but which is not difficult to model.*

   **Answer**:

   We originally built QG-Chem with the main scope of testing data assimilation algorithms on a system that reproduces the relatively fast (hourly) dynamics of reactive gases in the boundary layer. The longest DA experiment in the manuscript was of 3 days (length of the analysis plus forecast period in Section 4.2.3) with the model set-up reproducing the "Flux" experiment described in Crassier et al. (2000) on the central part of the QG-Chem domain. The experiment of Crassier et al. (2000) was meant to simulate the build-up of a typical summer pollution episode in Paris. Since time scales of dry deposition are typically longer [1], the authors also neglected this process in their study. Indeed, the experiments n. 1 and n. 3 (Section 4.2) of our manuscript were designed to examine the transient phase from a relatively unpolluted atmosphere to a locally polluted one. We do not agree completely with the reviewer about considering these conditions very unrealistic, because at hourly time scales the chemical system is most of the time out of the equilibrium due to the rapid variability of radiation, emissions and boundary layer height.

   However, we agree with the reviewer about the fact that over longer integration periods (several weeks), deposition terms become important in the chemical balances of trace gases, with time/space-averaged concentrations satisfying a chemical equilibrium between sources and sinks. Neglecting deposition in our original manuscript was mainly affecting the initial concentrations for the polluted experiment (n. 2), which started after a spin-up period of 20 days (instead of 1 day for
* * *
[1]If we consider for example a typical dry deposition velocity of $3.5 \cdot 10^{-3}$ m s$^{-1}$ for ozone over agricultural surfaces (Hardacre et al., 2015) and a boundary layer height (BLH) of 1.2 km, we obtain an e-folding time scale of about 4 days.

the experiments n. 1 and 3). It also limits the usability of QG-Chem for DA experiments that last longer than the ones presented in the manuscript. Therefore we decided to include dry deposition in the QG-Chem model and repeat all the experiments presented in the original manuscript. This lead, as expected, to different average concentrations for some species in case n. 2 and slightly different daily cycles in some cases, but all DA main results and conclusions remained the same. Details about the dry deposition mechanism are given now in the manuscript and some additional discussion of the differences between previous and new results can be found in detailed answers n. 20-21-24-25. We upgraded the manuscript with the new results.

2. *At some point, advection is neglected. This assumption is really too strong since it leads to a collection of 0/1-D box/column-models, whereas the purpose of this paper is more on the 4D-EnVar aspects where advection is critical.*

**Answer**:

The purpose of this study was twofold: i) presenting a new toy model that can be useful to test DA algorithms for atmospheric chemistry and ii) exploring the advantages of 4DEnVar method and a possible treatment of model errors. We agree on the fact that advection is a critical aspect to be examined in a DA testing framework. For this reason we considered a passive tracer in all experiments ($CO_2$) and we gave final conclusions based on several cycles of DA with both model error and advection, with changing wind field (Section 4.3, Table 6). However, aspects related to 4DEnVar and advection were already discussed in previous studies using meteorological models (Lorenc et al., 2015; Desroziers et al., 2014; Fairbairn et al., 2013). On the other hand, the performances of ensemble/variational DA on the very different dynamics of reactive gases are less documented in the literature. For example, DA issues related to the non-linearities of the chemical system were only recently pointed out by Tang et al. (2016). Some authors focused exclusively on 0-D box models to gain deeper understanding of the behavior of Ensemble Kalman filters (Tang et al., 2016) or chemical adjoint models (Hamer et al., 2015). It can be tricky to isolate possible issues within DA when the full system is simulated and chemical fields result from a complex interaction of all the underlying processes (chemistry, physics and transport). For this reason, we believe that a simplified set-up where only chemical processes are activated, can be a useful intermediate step when testing a novel technique, as for example the bias estimation procedure of this study. Finally, conditions of weak winds and local accumulation of pollutants can also occur in real cases, for which a 0-D model can provide useful insights. Therefore, we prefer to keep this experiment in the manuscript, which also shows some interesting effects of local emissions error.

In the revised manuscript, we better clarify the intent of the experiments without advection. The following sentences has been added in the introductory part of Section 4.2:

"The main objective of this section is to illustrate the application of the bias estimation procedure (Sec. 3.3) on chemical fields. Chemical interactions alone can already give rise to complex temporal dynamics, which can produce unattended behavior within typical hypotheses of most DA schemes (Tang et al., 2016). Therefore, a simplified model-set up is used in this section deactivating the advection of chemical species. This reduces QG-Chem to a collection of 0-D chemistry models and allows to focus on the effects of model errors on chemistry. Except from this, the same exact configuration

as before is used for 4DEnVar (initial condition error, ensemble size, localization). DA results in a more general case when both chemistry and advection are activated will follow in Sec. 4.3"

3. *The claim of not localizing in time is actually partially wrong; that is only an approximate statement. The dynamics of a consistent space-localization operator within the time window of 4D-EnVar was explored in Bocquet (2016), from which it can be understood that in the absence of time-localization the localization operator satisfies a Liouville equation.*

**Answer**:

We were aware of some recent studies that explored alternative approaches to apply a coherent localization operator in a 4D framework, like Bocquet (2016) or Desroziers et al. (2016). However, in this study we employed the approximation of a static covariance localization (Desroziers et al., 2014), which represented the simplest choice for the initial implementation of our toy model, it was already available in the DA library and provided satisfactory results with the DA windows used in our experiments (24 hours maximum). A deeper analysis of the impact of a static localization in our system and inclusion of more recent methodologies from latest studies will be considered in future work.

We slightly updated the text in Section 3.2 to avoid possible misunderstandings:

"This simplification, also called static localization, reduces significantly the numerical cost of the algorithm (Desroziers et al., 2014) at the price of degraded precision when increasing the length of the DA windows (Bocquet, 2016). The development of localization procedures that are more consistent with the dynamics of the forecast model is an ongoing research topic (Bocquet, 2016; Desroziers et al., 2016) and possible applications to QG-Chem will be considered in a future study"

4. *You are occulting the fact that the IEnKS has a fully nonlinear variational analysis which you don't have (this remark also applies to the current implementations of 4D-EnVar in meteorology). This is important because this could prove an asset when dealing with highly non-linear air quality models.*

**Answer**:

We agree with the reviewer about the importance of addressing nonlinear behavior of chemistry models within DA and we stated it several times in the manuscript (Pages 33 lines 1-3, Page 38 lines 25-26). We did not want to deliberately occult any information about the properties of IEnKS, which has been already proven to be effective in such conditions. Similarly to what was done for localization, we used here a basic set-up (one outer loop) to examine other aspects of chemical DA and left a thoughtful explorations of strong departures from the linearity for a future work.

The capacity of IEnKS to provide a fully non-linear analysis has been highlighted in the revised manuscript. The following lines are added in the introduction, when describing the IEnKS:

"A major asset of IEnKS for chemistry applications is that it can also account for strong non-linearities of the forecast model."

and at the end of Sec. 4.2.2:

"This difficulty could be overcome by introducing external loops within 4DEnVar, similarly to how it was already done with the IEnKS algorithm (Haussaire and Bocquet, 2016)."

5. *Actually 4D-EnVar should be compared to the EnKF. This is pretty obvious for assimilators using both ensemble filter and variational methods. Your standpoint looks like a biased one from specialists only used to variational methods. I do not encourage you to redo everything replacing 3D-Var with the EnKF. But several of your statements should be mitigated.*

**Answer**:

We acknowledge that a comparison with EnKF would have been probably more fair and provided more robustness to the conclusions of the study. However, previous studies already compared 4DVar and EnKF for air-quality applications and found similar results of the two schemes (Wu et al., 2008). Several state-of-the-art European air-quality operational models are based either on 3D-Var and EnKF and extensive comparisons of yearly chemical reanalyses showed very similar performances for both algorithms in practice (Rouil and the MACC team, 2014). For these reasons we considered that 3DVar represents a good-enough reference for evaluating the 4DEnVar scheme proposed in this study. Finally, implementing EnKF would have required a considerable amount of additional work and will be considered for a future study.

We included the following discussion at the end of Sec. 4.3:

"We finally remark that EnKF, which also takes advantage of the information available from the ensembles, could have represented an alternative and possibly more accurate reference than 3D-Var for this study. However, it is significantly more costly than 3D-Var, it introduces some difficulties like the definition of an optimal inflation procedure (Constantinescu et al., 2007a; Gaubert et al., 2014) and was not yet available in the OOPS DA library at the time of the study. A more comprehensive comparisons of DA schemes of increasing complexity and cost within QG-Chem is left for a future study, the present being focused especially on the bias correction procedure using 4DEnVar."

6. *From my understanding, the bias removal technique that you propose is actually a parametrized one since you pick up the perturbations that you apply to the ensemble. This is just a stochastic variant of the parameter estimation technique. If I am correct, you should substantially mitigate your claim on this point.*

**Answer**:

The proposed bias estimation is based on the implicit treatment of model errors done in the 4DEnVar state estimation. The analogy between 4DEnVar and the weak constraint 4D-Var (Desroziers et al., 2014) suggests that there is no particular limitation to the representation of model error that can be used. In the manuscript we analyzed the case of model errors that result from chemical emissions, which is a typical source of errors in chemistry models. Since we can identify emissions with a model parameter, the bias estimation has a strong analogy with the parameter estimation typically done within the EnKF framework. The main difference is that the bias estimation is done here only in the model state space,

without any update of model emissions (parameters). Therefore we agree with the reviewer on the fact that in some sense we "pick up the perturbations that we apply to the ensemble".

Nevertheless, the proposed method, not being linked to a particular parameter selection, should allow to correct biases as well as in more general cases where the ensemble is not generated by perturbing a particular set of parameters. We could think for example of stochastic ensemble perturbations or even multi-model ensembles (Marécal et al., 2015), where each model has its own chemical mechanism. Therefore, even if in certain cases the analogy might be strong, we prefer not to associate this method to a "parameter estimation technique", since fundamentally it does not address parameters estimation. Still, to obtain a reliable bias estimation, the model error dynamics has to be correctly represented by the ensemble, whatever the source of uncertainty (parametric or not). This is a crucial step that deserves further investigation using real air-quality models and observations, as already stated at page 38 lines 23-24.

7. *To shorten the paper, I would suggestion to get rid of the first configuration. It is unrealistic and I believe a bit detrimental to the paper anyway.*

   **Answer**:

   Since the conclusions of the DA experiments did not show significant differences among the configurations n. 1 and n. 2, we followed the suggestion of the reviewer and removed the first one from the manuscript (Sections 4.1 and 4.3) . However, we kept the model error experiment in section 4.2, which represents a pollution build-up case corresponding to the former configuration n. 1 (see also reply n. 2). The following text has been added in the introductory part of Section 4.2 to clarify this:

   "The main difference with the previous section is that experiments are done here during the spin-up period of the model (day 2 to 4). This allows to examine the model error estimation during the pollution build-up period, when the chemical system is in a transient phase and daily cycles of reactive gases are not yet stationary. This represents a more challenging and realistic situation for testing the bias correction procedure (Eq. 19), which is fully consistent only in stationary conditions."

8. *Please number all your equations. Avoiding excessive numbering is usually reserve for books, not for articles especially meant for peer-reviews.*

   **Answer**:

   All equations have been numbered in the revised manuscript.

**Additional remarks on the revised manuscript**

Doing again all the experiments after the inclusion of chemical deposition in QG-Chem let us discover two minor issues affecting the results presented in the original manuscript:

1. The correlation length of the initial perturbation used for the $CO_2$ experiments with a perfect model (Sec. 4.1) was erroneously set to 1000 km instead of 750 km as for the other species. This explained the different sensitivity of the $CO_2$

4DEnVar analysis to the size of the ensemble and the localization scale (Fig. 8 and 9 of the original manuscript), which was a bit surprising. New results (Fig. 5 and 6 of the revised manuscript) show now similar sensitivities for $CO_2$ as for other species. Therefore, the discussion dedicated to the $CO_2$ case in Sec. 4.1.3 (Page 26, lines 14-15 and 27) has been removed.

2. The ensemble of forecasts used to compute the 4DEnVar analyses with both the initial perturbation and model error enabled (Sec. 4.2, Fig. 11 bottom plot, Fig. 12 top right plot, Fig. 13, 14 and 15) were erroneously done with the advection enabled. This introduced an incoherence between the ensemble and the other simulations (truth, forecast, observations), which produced some biases in the model error estimation as well (e.g., the negative bias in the estimated model error in Fig. 11 and 12). The new results (Fig. 8, 9, 10, 11, 12 in the revised manuscript) are not anymore affected by this problem. The discussion has been revised where needed (e.g. Page 30, lines 19-24) and the color scale in Fig. 12 and 14 (Fig. 9 and 11 in the revised manuscript) has been modified to better display the positive model error values found for all experiments.

**2   Reply to specific comments**

1. *Page 1, line 2-3, "using a reduced-order chemical transport model based on quasi-geostrophic dynamics": this statement is ambiguous since we do not understand whether it is a CCMM or a CTM. Please clarify.*

   **Answer**:

   Even tough QG-Chem is a one-way coupled system, since the meteorology is not affected by the chemistry, we changed the sentence to "... using a reduced-order coupled chemistry-meteorology model based on quasi-geostrophic dynamics...". This should clarify the ambiguity. However, as already explained in the text our manuscript does not deal with aspects concerning the one-way coupling between meteorology and chemistry. A CTM-like configuration is actually adopted for the reported experiments. Aspects concerning coupled DA could be investigated in the future with QG-Chem.

2. *Page 1, line 4: "to a generic software library for data assimilation": which one? This meant to become a GMD paper. That type of info should be mentioned even in the abstract.*

   **Answer**:

   We changed the text with "to the software library for data assimilation OOPS (Object Oriented Prediction System)" . A reference to OOPS is now given in the manuscript.

3. *Page 1, line 12: "analysys" -> "analysis".*

   **Answer**: Done

4. *Page 1, line 12-13: "A comparison with results of 3D-Var, widely used in operational centers, shows that, for some species, analysys and next day forecast errors can be halved when model error is taken in account": A similar result was*

*obtained by Haussaire and Bocquet (2016). They showed that by using an ensemble forecast of the meteorology, thus partially accounting for model error, the root-mean-square error of IEnKS (a nonlinear 4D-EnVar method) on the low-order online tracer model L95-T was improved by 25% to 50%.*

**Answer**:

The suggested sentence and the corresponding reference is now added when discussing the reanalysis results of the repeated assimilation cycles (Sec. 4.3)

5. *Pages 3, line 6: "next day..." -> "Next day...".*

    **Answer**: Done

6. *Page 3, line 13-14: Haussaire and Bocquet (2016) also did a similar experiment simultaneously estimating gaseous concentrations and fluxes using a nonlinear 4D-EnVar.*

    **Answer**:

    We were aware of the above-mentioned study. However, in this paragraph, we cited only studies that used more standard algorithms (variational or EnKF) since the introduction of EnVar methods follows later. The reference to Haussaire and Bocquet (2016) concerning the application to the emissions estimation in a chemistry model has been reported again later in the introduction (see also reply to question n. 11.)

7. *Page 3, line 15: "Results seems promising but still relies on the assumption that the model is perfect, i.e. that there are no additional sources of uncertainties in the model forecast other than the controlled variables (i.e. the initial state and the selected emissions)": This is a rather biased statement. The model per see is not assumed perfect anymore, but only part of it, such as the dynamical part. This should be explained in a less biased way. If you estimate several parameters of the parametrization of a CTM, you are morally assuming that the model is imperfect. It is just that model error is parametrized.*

    **Answer**:

    We were referring in particular only to the two previously cited studies (Elbern et al., 2007; Hamer et al., 2015), which make use of 4D-Var in the strong constraint formulation. We agree with the reviewer about the fact that addressing parameter estimation, even with 4D-Var, corresponds morally to address in part the model error issue. However, this approach becomes rapidly unpractical when the possible sources of model error can invest the several thousands parameters of complex CTM (e.g. time and geographically varying emissions, photo-chemistry coefficients, deposition velocities, wind forcing fields etc.). In this sense we can probably state that the strong constraint 4D-Var can address air-quality DA within an "almost"-perfect model hypothesis. We revised the text in the manuscript in the following way:

    Results seems promising but still relies on the assumption that the model is *almost* perfect, i.e. that there are no additional sources of uncertainties in the model forecast other than the controlled variables (i.e. the initial state and the selected

emissions). This can lead to the over-correction of control variables when other non negligible model errors exist, for example due to the meteorological forcing, photo-chemistry coefficients, dry or wet deposition.

8. *Page 4, line 4-5: "The IEnKS (Bocquet and Sakov, 2014) is a fixed-lag ensemble Kalman smoother formulated under perfect model assumptions, which can also be used to estimate erroneous model parameters through an augmented state formalism (Haussaire and Bocquet, 2016)." No! the IEnKS is an iterative ensemble Kalman smoother. This is quite different from the standard "fixed-lag ensemble Kalman smoother"! It is better described as a nonlinear 4D-EnVar method. Please correct. For instance: "The iterative ensemble Kalman smoother (IEnKS, Bocquet and Sakov, 2014) is a nonlinear 4D-EnVar formulated under perfect model assumptions, which can also be used to estimate erroneous model parameters through an augmented state formalism (Haussaire and Bocquet, 2016)." Moreover, the augmented state formalism applied to the IEnKS was first demonstrated in Bocquet and Sakov (2013).*

**Answer**: The text has been revised according to the reviewer suggestion.

9. *Page 4, line 8-10: "These type of approaches are generally referred in the literature as ensemble-variational EnVar (Lorenc, 2013), as opposed to "hybrid" methods, which make use of ensembles only to specify error covariances matrices in variational algorithms (Belo Pereira and Berre, 2006)." - Instead of "These types of approaches..." it would be much better to write "These approaches", to avoid any ambiguity in the rightful claim that the IEnKS, the 4D- Var-EnKS and the 4D-EnVar are ensemble variational techniques. - What you call the "hybrid" methods are in fact called EDA standing for ensemble of data assimilation methods.*

**Answer**: Done

10. *Page 4, line 18: "the need in IEnKS to select a number of model parameters among all the possible erroneous parameters in complex CTMs": yes, but the EnKS and the IEnKS can account for stochastic perturbations in the integration step of the ensemble as already demonstrated in Haussaire and Bocquet (2016), section 3.2, configuration Offline 2. Please amend your statement.*

**Answer**: To avoid possible misunderstandings we removed this sentence from the manuscript.

11. *Page 4, line 23-25: "To the knowledge of the authors, EnVar type methods have not yet been implemented in air-quality or atmospheric chemistry models and no previous study has already examined the potential of 4DEnVar for chemical DA": Without any ambiguity, they have been tested in Haussaire and Bocquet (2016). I do not see any problem in recognizing that fact. Please correct.*

**Answer**:

We were originally referring to the 4DEnVar algorithm described in Desroziers et al. (2014). Given the conceptual similarity with IEnKS we modified this sentence as follows:

"To the knowledge of the authors, EnVar type methods have not yet been implemented in air-quality or atmospheric chemistry models and only one study has already examined the potential of EnVar methods for chemical DA (Haussaire and Bocquet, 2016)."

12. *Page 4, line 26: "Then" -> "Than".*

    **Answer**: Done

13. *Page 5, line 6: "This allow" -> "This allows".*

    **Answer**: Done

14. *Page 5, line 14: "under a generic library for data assimilation": Again, which one? Please be specific.*

    **Answer**: The reference to OOPS has been added.

15. *Page 6, line 18-19: "For all the experiments presented in this study a coarse resolution of 16x8 grid points has been used.": That is quite a low resolution and in contradiction with one of the early promises: "This choice permits to examine the behavior of DA in presence of complex gradients of wind fields and vorticity". Please revise or tune down your promises.*

16. *Page 6, line 21-22: "The only desired property is to obtain wind fields that exhibit typical patterns of the complex atmospheric circulation." which you don?t have with such a resolution. I am not criticizing your choice but the claims that are not matched by what you present. Please rephrase.*

    **Answer to 15 and 16**: We do not fully agree with the reviewer on this particular point. Within the QG model, typical patterns of Rossby waves can develop also with low horizontal resolutions, such as the ones that has been used in the study. The fields in Figure 1 of this document have been obtained changing only the spatial resolution of QG-Chem from the original 750 km (16 by 8 grid points) to 150 km (80 by 40 grid points). We can observe, that both meteorological and chemical fields are not significantly more complex than the ones that were used for the study (Figure 3 in the older manuscript, Figure 2 in the newer). We consider that the main requirement of reproducing vorticity fields that look like those typical of large scale circulation are satisfied also with the lower resolution. Higher resolutions would be beneficial to better examine aspects of DA such as the impact of the observational network, the spatial localization operator or interactions between small and large scale in chemistry. However, this was not the main objective of the current study.

17. *Page 7, line 5: "(Cariolle D, personal communication)": Daniel Cariolle is the third author. Remove this or give more details.*

    **Answer**:

    The reference has been removed. Some details of the method are given afterwards in the manuscript.

18. *Page 7, line 6: "which has a special treatment of the Jacobian matrix". This is too vague. Please be more specific.*

**Answer**:

A manuscript on the ASIS solver is currently in preparation. We recall here only the main characteristics of the ASIS method that are given later in the paragraph. We removed this too vague sentence.

19. *Page 7, line 13: "The meridional boundary conditions for chemical species are set to climatological values.": Ok, but what type of numerical boundary conditions? That is important.*

    **Answer**:

    The text has been revised as follows:

    The chemical field is set equal to climatological values on the N-S boundaries, which corresponds to Dirichlet type conditions.

20. *Page 7, line 14-15: "Moreover, no physical removal process for the chemical species has been included in the model so far.": This is both problematic (because this a key process of air pollution modeling) and odd (because it is not so really difficult to implement). Actually accounting for removal processes here is as important as having a fine, realistic meteorology. This is quite a weakness of your paper. The budget of all species is strongly affected. This also leads to an unbalanced photochemistry (induced by a wrong ratio of precursors).*

    **Answer**:

    With the inclusion of dry deposition mechanism in the revised manuscript, these lines are removed.

21. *Page 7, line 21: How does this configuration relate to regional air pollution modeling? Please elaborate.*

    **Answer**:

    Lines 25-27, page 7 have been replaced by:

    This configuration relates to regional air pollution modeling mainly concerning the type and amplitude of chemical emissions, spatial hetereogenity of sources and presence of boundaries.

22. *Page 8, Fig. 1: Did you show the grid? If not, could you please do so. That would help.*

    **Answer**:

    The model grid is now displayed in Figure 1 of the new manuscript.

23. *Page 10, table 3: Please give the extended name of each species in a another column.*

    **Answer**: Done in the revised manuscript

24. *Page 12, Figure 2: Haussaire and Bocquet (2016) have more realistic values of ozone for regional air pollution with a simplified CCMM than with your model! Please discuss this. Besides, the absence of a clear daily cycle for ozone is worrying and cast doubts.*

25. *Page 13, Figure 3: The magnitude and variation of ozone concentration is not realistic. I would have thought it should for such a toy-model.*

**Answer to 24 and 25**:

After the implementation of dry deposition in QG-Chem, the magnitude of the average ozone concentration in former Fig. 3 (Fig. 2 in the revised manuscript) is now closer to typical values encountered during summer periods in Europe. As discussed previously (see reply n. 1), the hourly variability was not much affected by the introduction of dry deposition. Figure 2 in this document shows that the typical variability of surface ozone in real models and observations is not too far from the one simulated with QG-Chem.

26. *Page 13, "by the variance:": You mean by the covariance matrix?*

**Answer**:

The text has been modified as follows: The covariance matrix $\mathbf{B}$ is modeled through the sequential application of 1D square-root correlation operators and a diagonal matrix, representing the background error standard deviation:

27. *Page 14, lines 3-9: What is the point in using the diffusion equation trick to obtain $\mathbf{C_x}$ for such toy-model and such 1D-correlation function? And why not for $\mathbf{C_y}$ ?*

28. *Page 14, lines 3-9: Why did you use a 2D correlation function C x,y ? There, using the diffusion equation would have been more meaningful(?).*

**Answer to 27 and 28**:

We built QG-Chem on top of the original operators that can be found in the OOPS distribution for the QG model. We have verified the code and found that the horizontal correlation operator $\mathbf{C_x}$ was actually not using a diffusion model. It is based on a Fast Fourier Transform approach to apply a filter in the spectral space. Since the domain is not periodic in N-S direction, $\mathbf{C}_y$ uses instead a symmetric positive-definite matrix filled with Gaussian correlations. We corrected the text as follows:

$\mathbf{C}_x$ and $\mathbf{C}_y$ are isotropic homogeneous correlation operators providing Gaussian spatial structures.

29. *Page 14, line 25: "The 4DEnVar algorithm is meant to solve the main drawbacks of 3D-Var": Of course that is not its primary purpose. Please replace "is meant to" with "can".*

**Answer**: Done

30. *Page 14, line 28: "written as :" -> "written as:"*

**Answer**: Done

31. *Page 14, 31-32: "The cost function is computed for an assimilation window that can span several hours or days": How long is the data assimilation window in your study? That is a key value that must be mentioned and discussed, including in the paper at this point.*

**Answer**:

The main parameters of the 4DEnVar specific to our study are discussed later in the manuscript (Sec. 4). In particular the choice of the assimilation windows and sub-windows was discussed at page 18, lines 7-11. Since we wanted to provide here only a general description of the DA method, we removed also the sentence concerning the length of the sub-windows.

32. *Page 15, line 10: "an hybrid 3D-Var": What is an hybrid 3D-Var for you? What you have described does not look like what is usually understood as an hybrid algorithm.*

    **Answer**:

    We consider a 3D-Var to be hybrid when the background error covariance is a weighted sum of a static and an ensemble derived term (Wang et al., 2013). At line 10 we described a particular case that corresponds to a zero weight on the static term. We think that this could still be a particular case of an hybrid 3D-Var. However, to avoid possible misunderstandings, we decided to remove the sentence.

33. *Page 15, line 29: "...no time localization is applied and the same 3D (and multivariate) correlation operator C is used for all 4DEnVar sub-windows": As I explained, this is not really a "no localization" condition. Please mention this.*

    **Answer**:

    The requested clarification has been made (see reply n. 3) and "no time localization is applied" has been removed.

34. *Page 15, line 32: "Hence, for the experiments presented in this study, we could use the covariance operator described in (6) by setting the variance terms to one.": The sentence is confusing. I would write "Hence, in order to specify C, we could use the covariance operator described in (6) by setting the variance terms to one."*

    **Answer**: The sentence has been modified according to the suggestion.

35. *Page 16, line 1: "is an ongoing research topic (Bocquet et al., 2015)" should be "is an ongoing research topic (Bocquet, 2016)"*

    **Answer**: Done. A second recent reference has also been included (Desroziers et al., 2016).

36. *Page 16, lines 19-23: There is an approximation here. These are not strict inequalities. You should use at least one $\simeq$ symbol.*

37. *Page 16, line 26: Again, you should use the $\simeq$ symbol here.*

    **Answer**: The approximate equality symbol is used now whenever linearization is used.

38. *Page 17, before section 4: How are the perturbations generated? This is a critical part of the EnVar schemes, rigorously derived in the IEnKS (Bocquet and Sakov, 2014), and approximately so in other EnVar systems (so far).*

    **Answer**:

The choice of the perturbations used in the numerical experiments is already detailed in the results section (Sec. 4.1 and 4.2).

39. *Page 18, line 1: "and by adding a normally distributed error": What did you do with the negative measurements?*

    **Answer**:

    Negative observations did not occur in the numerical experiments, since the used standard deviation was significantly smaller than the truth concentrations. Other error distributions (e.g. log-normal) could probably be adopted when exploring cases with larger departures from the truth.

40. *Page 18, line 5-6: "The meteorology is never observed neither perturbed.": so this is a CTM-like experiment. This should be emphasized since this is critical.*

    **Answer**:

    We clarify this also in the introductory paragraph of section 2 by adding the following sentence:

    Aspects of DA concerning the coupling between meteorology and chemistry are also left for future work, with the present using QG-Chem in a CTM-like mode.

    and extending the previous one to:

    The meteorology is never observed nor perturbed, which corresponds to use QG-Chem in a CTM-like mode.

41. *Page 18, line 22: "gaussian" -> "Gaussian".*

    **Answer**: Done

42. *Page 18, line 22: 1 in the Gaussian distribution should actually be the identity matrix.*

    **Answer**: The formula has been updated.

43. *Page 19, line 1: "ensemble localization" -> "localization of covariances".*

    **Answer**: Done

44. *Page 19, Table 4: That would be good to have the relative value of the stds, i.e. divided by a standard concentration value.*

    **Answer**:

    We added the relative value as percentage of the average chemical concentrations given in Figure 3 (Figure 2 in the revised manuscript).

45. *Page 19, line 5: "All vertical terms of the covariance or localization matrix are always set to zero in this study, since only the bottom layer of the QG-Chem model is considered": I do not understand the justification. Could you please clarify?*

**Answer**:

Even if QG-Chem has two vertical layers, in this study we limited our attention to the bottom layer, to focus on boundary layer chemistry and 2D transport. Moreover, there is no chemical coupling between the two layers and no vertical transport of chemical species. Assimilated observations are also located in the bottom layer and the observation operator does not interpolate between levels. Therefore, setting the vertical correlation to zero in this study has only two effects: i) it provides vertically uncorrelated errors on the initial condition and ii) it prevents spurious 4DEnVar increments to spread in the top layer. Anyhow, this does not have any influence on the presented results, which are only displayed only for the bottom layer.

The sentence has been modified as follows for sake of clarity:

Since we use QG-Chem in a CTM-like configuration and the two layers are chemically not coupled, the vertical terms of the covariance or localization matrix are always set to zero in this study, without any impact on the presented results.

46. *Page 19, line 11: "Fig. 4 " -> "Figure 4".*

    **Answer**: Done

47. *Page 19, line 20-21: "The memory of the initial condition is rapidly lost for O3, as it was also demonstrated within regional air-quality models (Jaumouillé et al., 2012).": I doubt Jaumouillé et al. (2012) were the first to show/discuss this. Please give an earlier reference in addition to yours unless I am mistaken.*

    **Answer**:

    The following reference is also given: Wu et al. (2008)

48. *Page 20, line 4: "...which stays...": what is "which" referring to? Please clarify.*

    **Answer**:

    Since the comparison of chemical trajectories between experiments 1 and 2 is not presented anymore in the revised manuscript, lines 2-4, Page 20 have been removed.

49. *Page 21, line 1: "Fig. 5 a,d Fig. 7" -> "Figures 4 and 7".*

    **Answer**: Done

50. *Page 21, line 5: This definition is a bit confusion since Eq.(17) has a normalization and Eq.(18) has not. Please clarify or use non-confusing notations.*

    **Answer**:

    To improve the readability of the results we provided values of the RMSE gain of DA always as percentage of the average chemical concentration. Equation (18) is only used to define the RMSE (in absolute value) that is used to derive the RMSE gain (Eq. 17). The clarification has been included in the revised manuscript.

51. *Page 24, line 10: "satisfactory" is inappropriate. Be more specific. Per se the RMSE do not mean much because of what you wrote. Only the comparison of the RMSEs between the 3D-Var and 4D-EnVar is relevant. This comparison yields satisfactory results.*

    **Answer**:

    We clarified the sentence as follows:

    The appearance of local but relatively small RMSE degradation can be tolerated in atmospheric chemistry, because the chemical system has a dissipative behavior and errors on the model state cannot grow during the forecast step.

52. *Page 25, line 25: "If this is not the case, a larger ensemble size allows in principle less severe localization (Ménétrier et al., 2015).": The reference to Ménétrier et al., 2015 is inappropriate here (fully justified later on). This is very well known in ensemble data assimilation (especially for the EnKF) for 15 years. Cite an EnKF paper instead, or nothing since this is common knowledge.*

    **Answer**:

    The citation has been removed.

53. *Page 25, line 33: "The ensemble size is the main limiting factor in operational forecast centers...": This is not always true. For instance in Meteorology Environment and Climate Change Canada is using large ensembles, an option that they prefer (to better deal with inflation and localization).*

    **Answer**:

    The sentence has been changed to:

    The ensemble size can be one of the main limiting factors in operational forecast centers...

54. *Page 26, line 16: "In few cases..." -> "In a few cases...".*

    **Answer**: Done

55. *Page 26, line 21-22: "We remind that the objective of this study is to demonstrate the applicability of a DA algorithm that outperforms currently implemented methods in operational centers..." A biased statement since other ensemble-based methods are ignored.*

    **Answer**:

    The sentence has been modified as follows:

    We remind that the objective of this study is to demonstrate the applicability of a DA algorithm that could outperform currently implemented methods in operational centers...

    The same has been done at Page 5, line 12.

56. *Page 27, Fig. 8: Unfortunately the average RMSE of case 2, which I consider as the most enlightening indicator in this figure cannot be seen very easily because of the larger bars of the maximum. I suggest that you multiply by 5 the average values or, alternatively, provide a second y-axis on the right.*

    **Answer**:

    The average gain has been multiplied by 10 in the revised figures.

57. *Page 28, line 4: "...i.e. a case that cannot be addressed using 3D-Var or strong constraint 4D-Var." Indeed, but you could address it with an EnKF, and it has already been. Please mitigate your statement.*

    **Answer**:

    The sentence has been modified as follows:

    However, the main interest of 4DEnVar for atmospheric chemistry arises when the model is not perfect, i.e. a case that is more easily addressed using ensemble-based methods.

58. *Page 29, line 4: "A multiplicative factor of 2.35 has been sampled for the NO emissions of the forecast simulation...": I do not understand "has been sampled"; please clarify.*

    **Answer**:

    The has been modified as follows:

    A multiplicative factor of 2.35 has been used for the forecast simulation

59. *Page 29, line 5-6: "The advection has been deactivated in this set of experiments to better focus on the impact of emissions uncertainty on chemistry.": This seems too strong an assumption to me! the model becoming 1D on the vertical. Some of your conclusions are based on these experiments which have a limited scope because of this assumption. Advection is of course critical for qualitatively realistic atmospheric chemistry modeling.*

    **Answer**: See reply n. 2

60. *Page 32, line 12: "during night" -> "during the night".*

    **Answer**: Done

61. *Page 33, Fig. 13: the useful subtitles of each panel, as seen in Fig. 11, have disappeared. Please add them.*

    **Answer**:

    Results from only one experiment and different chemical species are shown in Fig. 13 (Fig. 10 in the revised manuscript), whereas 3 different experiments were displayed in Fig. 11 (Fig. 8 in the revised manuscript). For this reason no subtitles were given. We revised the legend of the figure as follows:

    Multivariate effective model error estimation. Temporal trajectories for day 2, as in Fig. 8 but for $NO_2$, NO, CO species (not assimilated), obtained assimilating only $O_3$ . Uncertainties were both in the $O_3$ initial condition and in the forecast

model (NO emissions), corresponding to the third experiment in Fig. 8. Only difference with the previous experiment in is that a non-zero multivariate localization coefficient is used here.

62. *Page 33, line 3: "This difficulty could be overcome by introducing external loops within 4DEnVar": this has been partly addressed in Haussaire and Bocquet (2016). Please mention it.*

**Answer**:

The reference has been included

63. *Page 36, line 1: The title of section 4.3 "Validation on multiple DA cycles" is very misleading, since you are not cycling the scheme, but only repeating one-cycle experiment, i.e. gathering statistics. Please rephrase.*

**Answer**:

The title of the section has been changed to:

Statistical comparison between 4DEnVar and 3D-Var

64. *Page 38, line 6: "The justification of using an ensemble method for operational air-quality DA, which is significantly more costly than 3DVar, has been demonstrated when model errors were introduced": This has been demonstrated long before by many teams using RRSQRT, EnKF for air quality. Please rephrase.*

**Answer**:

The sentence has been rephrased as follows:

"Clear advantages of using an ensemble method, which is significantly more costly than 3DVar, have been found when model errors were introduced"

This mitigates the original statement. However, to the knowledge of the authors, if we limit us to studies using real observations, only two studies found reduced RMSE in ozone forecasts when including parametric model errors in DA experiments (Constantinescu et al., 2007b; Tang et al., 2011). Moreover, results of Tang et al. (2011) were somehow mitigated by the fact that better ozone forecasts corresponded to worse $NO_x$ ones (Tang et al., 2016). Studies that strictly compared strong constraint 4D-Var and ensemble methods (mostly variants of EnKF) with the same chemical model and without estimating model errors did not show significant advantages of using ensembles (Skachko et al., 2016, 2014; Wu et al., 2008; Constantinescu et al., 2007a). We also did not find other studies than Jaumouillé et al. (2012) which compare ensemble based methods to the quite cheap 3D-Var scheme with static error covariances. Therefore, we think that additional comparisons are needed to draw robust conclusions on the delicate performance to cost ratio in chemical DA, using both realistic and simplified air-quality models.

65. *Page 38, line 7-16: Many of the statements there should be mitigated: systematically recall that these results have been obtained in the context of a simplified model (with unrealistic features). Generalized statements for 4D-EnVar cannot be made.*

**Answer**:

We already recall that these results have been obtained in the context of a simplified model few lines later (Lines 23-27) and that application to DA experiments using real observations is needed. Within lines 7-16 we just summarize the results that have been obtained in this study. In our opinion, this makes already clear that the given statements are not generalized to all possible applications of 4DEnVar.

66. *Page 38, line 15: "...on multiple cycles of DA...": this is very misleading, as discussed before! Strictly speaking this is wrong. Please remove this statement.*

    **Answer**:

    We changed the phrase to:

    This has been tested with success on several independent DA windows ...

67. *Page 38, line 18: "We conclude that 4DEnVar provides a practical and powerful algorithm for chemical DA.": Again, you have to mitigate this statement. "In the context of a low-order/simplified model, we conclude". Also powerful is too much.*

    **Answer**:

    The sentence has been changed to:

    We conclude that 4DEnVar is potentially of high interest for chemical DA.

68. *Page 38, line 26: "QG-Chem will represent a useful tool for this type of studies." -> "QG-Chem could represent a useful tool for this type of studies."*

    **Answer**: Done

[Figure]

**Figure 1.** Meteorological and chemical fields of QG-Chem (with deposition included) on day 20 computed using an horizontal grid of 80 x 40 points (resolution of ~ 150 km). Time-averaged fields for a 24 hours period on the left, time series of domain-averaged values for the same 24 hours period on the right. The wind field and the concentration of the four chemical species of interest (CO, $NO_2$, $O_3$, $CO_2$ not shown since constant and equal to 310 ppmv) are shown from top to bottom.

[Figure]

**Figure 2.** Top: average daily cycle of surface ozone during the period from 5-7-2010 to 21-7-2010 at European AIRBASE measurement sites. Measured ozone in black (averaged over 254 measurement sites), corresponding simulated ozone from two French operational air-quality models in red (CHIMERE) and green (MOCAGE). Bottom: average ozone modeled with QG-Chem on day 20 (corresponding to bottom-right plot in figure 2 of the revised manuscript, with deposition included)

---

## Author Comment (AC2) · 15 Sep 2016

**Accounting for model error in air-quality forecasts: an application of 4DEnVar to the assimilation of atmospheric composition using QG-Chem 1.0. Reply to referee # 2**

Emanuele Emili1, Selime Gürol1, and Daniel Cariolle1 1CECI UMR5318 CNRS/CERFACS, Toulouse, France *Correspondence to:* Emili (emili@cerfacs.fr)

**1 Reply to general comments**

We thank the reviewer for his comments, which helped to improve the manuscript. The detailed replies follow:

- 1. It seems that are tests are done setting to zero inter-species error correlation. While some justification is provided for this, perhaps an actual figure would help. For example on p. 23 it is said: Finally, 4DEnVar was capable to provide same good results as in Sec 4.1.1 when enabling the cross-variables covariances?. Results are not shown to back this
  - Answer:

statement?

We omitted showing the corresponding Figure since differences were not very significant. For completeness we report here in Fig. 1 the results (RMSE gain) obtained when assimilating all species at once and enabling cross-variables covariances, compared to those already shown in the original manuscript (univariate and independent experiments for each species). We note that differences are quite small. Moreover, summarized scores for both multivariate and univariate DA were already presented in Fig. 10 (Fig. 7 of the revised manuscript), when discussing the impact of multivariate localization. We removed the word "Finally" from the beginning of the sentence to avoid possible misunderstandings, since we just refer there to the previously discussed results (Page 25, line 7).

15 2. It is said that "Only surface observations are considered in this study" (p13, L28). There is a need to acknowledge as well the lack of vertical propagation of information from these surface variables in the context of the "toy" system. This represents a significant challenge in a real system.

**Answer:**

20

5

10

We agree with the reviewer about the importance of correctly propagating vertical information in chemical DA, especially for the assimilation of satellite data. QG-Chem could permit vertical propagation of information in DA experiments but it has an over-simplified vertical structure (two layers). Therefore, we think that vertical aspects of chemical DA remain beyond the scope of QG-Chem. We added the following sentence to the conclusions of the revised manuscript:

---

## Author Comment (AC3) · 15 Sep 2016

**Accounting for model error in air-quality forecasts: an application of 4DEnVar to the assimilation of atmospheric composition using QG-Chem 1.0 . Reply to referee # 3**

Emanuele Emili[1], Selime Gürol[1], and Daniel Cariolle[1]

[1]CECI UMR5318 CNRS/CERFACS, Toulouse, France

*Correspondence to:* Emili (emili@cerfacs.fr)

**1  Reply to general comments**

We thank the reviewer for his comments, which helped to improve the manuscript. The detailed replies follow:

1. *I found the last paragraph in section 2.1 to be unclear and somewhat confusing. For instance, it is not clear what was meant by ?The sources of the QG model provided by ECMWF...?. Do the authors mean that the code for the dynamical model was given to them by ECMWF? There are some further examples listed below. I therefore recommend that the authors check this text carefully for its clarity and re-write it.*

   **Answer**: Yes, the reviewer's interpretation is rigth. The text has been modified as follows:

   The code of the QG model that is distributed with the OOPS DA library have been used for this study (Y. Trémolet, personal communication).

2. *"The boundary conditions are taken cyclic...", but it is not clear what "cyclic" means in this context.*

   **Answer**: The text has been modified as follows:

   The domain is cyclic in the East-West direction, i.e. the model fields are periodic in this direction. The stream function is set to climatological values at meridional walls (Dirichlet boundary conditions).

3. *"For all the experiments presented in this study a coarse resolution of 16x8 grid points has been used." I have assumed that this is the horizontal resolution, but it is not clear whether 16 refers to the North-South dimension or the East-West. The authors should also write the horizontal resolution in terms of spatial resolution in km.*

   **Answer**: The sentence has been modified as follows:

   For all the experiments presented in this study a coarse resolution of approximately 750 km (16x8 grid points, respectively, for the East-West and North-South directions) has been used.

4. *Figure 1 would probably be improved with an explanation of what the grid lines represent. One presumes these are the outlines of the grid boxes used in the model, but the authors should state this clearly.*

**Answer**: In the original manuscript the displayed lines did not correspond to the model grid but were only used for spatial reference. In the revised manuscript all figures showing the model fields have been revised and now display the full model grid. An explanation has been added as well in the legend of Figure 1.

5. *Page 24, lines 1-2. The authors should probably remind readers that only the initial conditions are perturbed during these tests, and therefore NO2 converges to the truth due to the shorter lifetime of NO2 combined with the fact that there are no errors in the emissions.*

    **Answer**: The following line has been added to remind the reader:

    This happens because, when the model is perfect, the $NO_2$ field rapidly converges to the truth after few hours.

6. *Page 25, line 15. Can the authors think of any specific reasons why ozone and NO2 show similar behaviour to CO and CO2?*

    **Answer**:

    We think that such a behavior has been observed because the initial perturbation of $NO_2$, which is 16% of the average $NO_2$ concentration ($\sim$0.1 ppbv), is applied at midnight and is quite small compared to the amount of $NO_2$ that is produced hourly in our experiments as a consequence of NO emissions ($\sim$0.15 ppbv per hour). Hence, the amount of $NO_2$ available for photochemical reactions and ozone production later in the day is very slightly influenced by the initial perturbation. Cases with larger initial $NO_2$ perturbations, especially if applied during the active photochemical phase, could show different sensitivities to the initial condition. Additional experiments with a larger variety of initial perturbations should probably be performed to check if the conclusions given in this section remain valid. We included the following sentence in the manuscript to highlight this point:

    A possible reason is that the amount of $NO_2$ produced hourly by the oxydation of emitted NO is much larger than the applied initial perturbation. Therefore, the $O_3$ photochemical production, which happens later during the day, is not much influenced by the perturbation of $NO_2$ at midnight. It would be interesting to verify if similar results also hold when larger perturbations are applied during day-time. A wider exploration of different chemical regimes is left for a future study.

7. *The explanations for the emission perturbations at the beginning of Section 4.2 could benefit from slightly more explanation. It is not stated directly, but I assume from the existing text that the perturbations result in reductions in NO emissions. This should be stated directly. Also, does the size of the perturbations vary with time, and if so at what frequency?*

    **Answer**:

    The information was given at Page 29, line 4 but was actually not very clear. We updated the manuscript with the following sentence:

Forecast emissions are increased by a multiplicative factor of 2.35, whereas the log-normal distribution has been used to generate emission perturbations for the ensemble of forecasts. The emissions perturbation is constant in time but not in space, due to the geographical variability of emission factors (Fig. 1).

8. *Please can the authors add some further text to the Figure 12 caption to explain what the red and blue colours represent.*

5   **Answer**: The color bar has been modified in the revised manuscript (c.f. reply to reviewer n. 1, additional remarks)

All further technical comments have been considered for the revised manuscript.